# A Stochastic Polynomial Expansion for Uncertainty Propagation through Networks

## Abstract

Network-based machine learning constructs are becoming more prevalent in sensing and decision-making systems. As these systems are implemented in safety-critical environments such as pedestrian detection and power management, it is crucial to evaluate confidence in their decisions. At the heart of this problem is a need to understand and characterize how errors at the input of networks become progressively expanded or contracted as signals move through layers, especially in light of the non-trivial nonlinearities manifest throughout modern machine learning architectures. When sampling methods become expensive due to network size or complexity, approximation is needed and popular methods include Jacobian (first order Taylor) linearization and stochastic linearization. However, despite computational tractability, the accuracy of these methods can break down in situations with moderate to high input uncertainty. Here, we present a generalized method of propagating variational multivariate Gaussian distributions through neural networks. We propose a modified Taylor expansion function for nonlinear transformation of Gaussian distributions, with an additional approximation in which the polynomial terms act on independent Gaussian random variables (which are identically distributed). With these approximated higher order terms (HOTs), we obtain significantly more accurate estimation of layer-wise distributions. Despite the introduction of the HOTs, this method can propagate a full covariance matrix with a complexity of $O(n^2)$ (and $O(n)$ if only propagating marginal variance), comparable to Jacobian linearization. Thus, our method finds a balance between efficiency and accuracy. We derived the closed form solutions for this approximate Stochastic Taylor expansion for seven commonly used nonlinearities and verified the effectiveness of our method in deep residual neural networks. This general method can be integrated into use-cases such as Kalman filtering, adversarial training, and variational learning.

## 1 Introduction

A fundamental problem in uncertainty estimation and verification is to characterize how a given input distribution becomes transformed by the operant function (succinctly, $Y = f(X)$). When $f$ takes the form of a modern machine learning (ML) architecture, this problem quickly becomes analytically intractable, necessitating either sampling methods or approximation. Two dominant approximation techniques are Jacobian linearization (JL), i.e., deterministic first order Taylor expansion around the mean of input distribution, and stochastic linearization (SL), which uses expected value of the first derivative as gain and mean of output as bias, or $\hat{Y} = \mathbb{E}[\nabla_x f(X)] \circ (X - \mu_x) + \mathbb{E}[f(X)]$. Stochastic linearization minimizes mean square of the residual (Booton (1953); Kazakov (1954)), and has been used in the context of feedback control systems (see, e.g., Ching et al. (2010) or Elishakoff and Crandall (2017)). To give a few examples in ML contexts, Gandhi et al. (2018) Dera et al. (2021) Petersen et al. (2024) used Jacobian linearization for uncertainty propagation through networks, and Beiu et al. (1994); Abdelaziz et al. (2015) used a piece-wise linear approximation of **sigmoid** functions prior to JL. In Zhen et al. (2021), the authors used SL to provide an output covariance that enabled the determination of a certified covariance radius.

Perhaps unsurprisingly, these methods work best in low-uncertainty regimes. In high uncertainty setting, variance can arise not only measurement error (e.g. sensor noise in vision networks), but also the parameters

of the neural network itself. For example, in 2D batch normalization layers, activations are normalized by the square root of pre-trained variance, which is typically far smaller than 1 if input images are standardized. This normalization can easily drive variance of 'neuron'-level representations tenfold or more (Fig. 1).

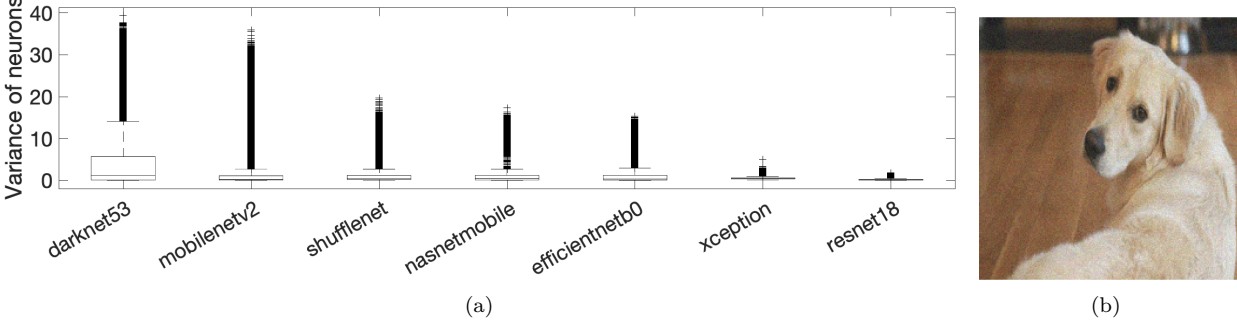

(a)                                                                     (b)

Figure 1: We survey preceding layers of nonlinearity in some pretrained image-classification networks (Redmon; Sandler et al. (2018); Zhang et al. (2017); Zoph et al. (2017); Tan and Le (2019); Chollet (2017); He et al. (2016)), and show how high the variance can be despite the moderate image noise (see (b)). (a) Boxplot of variance of neurons from one layer immediately followed by nonlinearity, and it is common to have high uncertainty propagating through nonlinear layers. These values are obtained by sampling 100 noisy images. (b) Input image injected by Gaussian noises $N(0, \sigma^2 = 100)$. Note that due to the normalization in the ImageInputLayer (e.g. z-score), the equivalent (after the ImageInputLayer) pixel-wise variance ranges from 0.0015 to 0.0301. Thus, normalizing input images does not mitigate this effect. This is consistent with previous findings that many models were not robust to even negligible noises on input (Szegedy et al. (2013); Biggio et al. (2013); Carlini and Wagner (2017); Ilyas et al. (2018)).

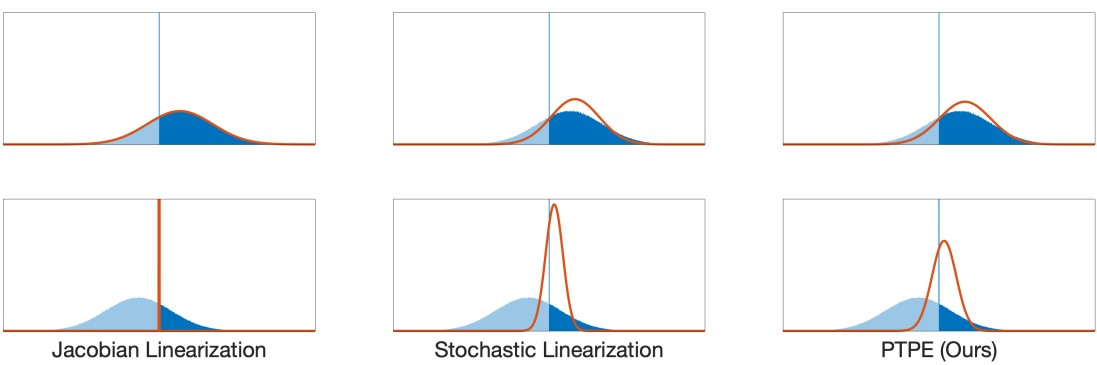

Figure 2: Blue: Two Gaussian distributions, with positive (top) and negative (bottom) mean, propagated through the ReLU function. Orange: Approximations of the output distribution suggested by three uncertainty propagation methods.

Another approach, the method of moments, aims to obtain the first and second moments of propagated distributions, and then use a Gaussian distribution with the same moments to approximate the output distribution. Some early works include Frey and Hinton (1999) for ReLU, Spiegelhalter and Lauritzen (1990) and MacKay (1992) for **sigmoid** and **tanh** function using two different approximations. Subsequently, Hernández-Lobato and Adams (2015) Wang et al. (2016) Gast and Roth (2018) explored the usage of this method for training probabilistic neural networks, and Wang and Manning (2013) in drop-out training. The work Shekhovtsov and Flach (2019) also developed an analytic approximation of propagating uncertainty through argmax and softmax under the assumption of logistic and Gumbel priors. Amidst these studies, a critical issue surfaces: the derivation typically depends on the assumption of independent Gaussian input. As

we will later confirm, this assumption result in significant approximation error when correlations between neurons are substantial, for example, in convolution neural networks with overlapping convolution kernels. In Shridhar et al. (2019), the authors attempted to address this issue by training a hyperparameter for each convolution kernel to compensate for missing covariance, but this method can only be optimized on averaged rather than case-by-case, and requires extra training data perturbed by only a narrow band of noise.

**Contributions** To summarize, current uncertainty propagation of variational Gaussian distributions through nonlinear layers relies on either (1) linearizing the nonlinearities or (2) direct derivation (or approximation) of the first two moments of output distributions under the assumption of zero correlation. The former allows propagating full covariance matrices, but introduce errors from ignoring higher order moments' contribution to covariance calculation; the latter introduce errors from ignoring correlation' contributions to variance calculation. If deterministic first-order Taylor expansion is used, additional error is introduced from ignoring the uncertainty of the first derivative.

To address these shortcomings, we here postulate a generalized framework of using a stochastic polynomial expansion as a surrogate nonlinearity, and derived closed-form solutions of the mean, covariance, and cross-covariance of propagated multi-variate distributions, for seven nonlinearities that are ubiquitous in modern ML network constructs. This is achieved with a computational complexity of $\mathcal{O}(n^2)$, comparable to that of first-order Taylor expansion.

**Relation to Bayesian Learning for Neural Networks** In exact Bayesian inference, one needs to solve the posterior distribution of the network parameters $\theta$ given data $D$, i.e. $p(\theta|D) = \frac{p(D|\theta)p(\theta)}{\int p(D|\theta')p(\theta')d\theta'}$.
The marginal distribution is an integral of the likelihood $p(D|\theta')$ over all possible combinations of network parameters. Since the likelihood is intractable due to the nonlinearity, and sampling methods are not practical in the limit of large parameter size, this integral has to be approximated. Approximating this likelihood with variational Gaussian distributions has been previously explored Hinton and van Camp (1993); Graves (2011); Hernández-Lobato and Adams (2015); Wu et al. (2019). Often, deterministic first-order Taylor expansion is used for layer-wise propagation of Gaussian distributions. Thus, our method may contribute to higher accuracy in such a Bayesian inference setting.

## 2 Theory

Let $\mathbf{X} := (X_1, X_2, \cdots, X_n)^\top$ be a Gaussian random vector following $\mathcal{N}(\tilde{\boldsymbol{\mu}}, \tilde{\boldsymbol{\Sigma}})$, and $\boldsymbol{\Xi} := \mathbf{X} - \mathbb{E}[\mathbf{X}]$. Let $\bar{f}(\cdot)$ be a smooth, univariate function. We define the vector function $f(\mathbf{X}) = (\bar{f}(X_1), \bar{f}(X_2), \cdots, \bar{f}(X_n))^\top$. Here, we have in mind that $f(\cdot)$ describes the activation function at the output of a feedforward layer. Now, let us define a set of i.i.d. surrogate distributions $\boldsymbol{\Xi}_{(s)} \sim \mathcal{N}(\mathbf{0}, \tilde{\boldsymbol{\Sigma}})$, $s = 1, 2, ..., S$.

We now propose the central construct in this paper, the **pseudo-Taylor polynomial expansion (PTPE)** of $f(X)$ as:

$$g(\mathbf{X}) = \mathbb{E}\left[f(\mathbf{X})\right] + \sum_{s=1}^{S} \frac{\mathbb{E}\left[\nabla_{\mathbf{x}}^s f(\mathbf{X})\right]}{s!} \circ \left(\boldsymbol{\Xi}_{(s)}^{\circ s} - \mathbb{E}\left[\boldsymbol{\Xi}_{(s)}^{\circ s}\right]\right) \tag{1}$$

We postulate and later show that this expansion provides a tractable and accurate approximation of $f(X)$ for the purposes of propagating uncertainty through feedforward network architectures.

In the above, we use the form of a Taylor polynomial expansion to describe the behavior (in expectation) of the function $f(\mathbf{X})$ subject to the stochastic input $\mathbf{X}$. The choice of the i.i.d. surrogate polynomials, $\{\mathbf{1}, \boldsymbol{\Xi}_{(1)}, \boldsymbol{\Xi}_{(2)}^{\circ 2}, \boldsymbol{\Xi}_{(3)}^{\circ 3}, \cdots\}$, is made to simplify the ensuing derivations. Note that if taking only the first two terms, this expansion is equivalent to stochastic linearization, because $\boldsymbol{\Xi}_{(1)} - \mathbb{E}[\boldsymbol{\Xi}_{(1)}]$ is equivalent to $\mathbf{X} - \mathbb{E}[\mathbf{X}]$. It is straightforward to observe that $g(\mathbf{X})$ has the same first moment as $f(\mathbf{X})$ because all terms after the first one are designed to have zero mean. In the following, we will provide empirical evidence that the second moment is well-captured for many common activation functions.

First, we derive the solution for covariance and cross-covariance using the proposed stochastic polynomial expansion.

**Lemma 1.** *Define*

$$\boldsymbol{A}_0 = \mathbb{E}\left[f(\mathbf{X})\right] \qquad\qquad \boldsymbol{A}_1 = \frac{\mathbb{E}\left[\nabla_{\mathbf{x}} f(\mathbf{X})\right]}{1!} \qquad\qquad \boldsymbol{A}_2 = \frac{\mathbb{E}\left[\nabla_{\mathbf{x}}^2 f(\mathbf{X})\right]}{2!} \qquad\qquad \cdots$$

*Then, the covariance matrix of $g(\mathbf{X})$ is*

$$\Sigma_{g(\mathbf{X})} = \sum_{s=1}^{S} \boldsymbol{A}_s \circ \left( \mathbb{E}\left[\boldsymbol{\Xi}_{(s)}^{\circ s}\ \boldsymbol{\Xi}_{(s)}^{\circ s\top}\right] - \mathbb{E}\left[\boldsymbol{\Xi}_{(s)}^{\circ s}\right] \mathbb{E}\left[\boldsymbol{\Xi}_{(s)}^{\circ s}\right]^{\top} \right) \circ \boldsymbol{A}_s^{\top} \tag{2}$$

*for an $S$-th order expansion. For $S = 3$,*

$$\begin{aligned}
\Sigma_{g(\mathbf{X})} = & \boldsymbol{A}_1 \circ \tilde{\boldsymbol{\Sigma}} \circ \boldsymbol{A}_1^{\top} + \\
& \boldsymbol{A}_2 \circ \left(2\tilde{\boldsymbol{\Sigma}}^{\circ 2}\right) \circ \boldsymbol{A}_2^{\top} + \\
& \boldsymbol{A}_3 \circ \left[6\tilde{\boldsymbol{\Sigma}}^{\circ 3} + 9\ \mathrm{diag}(\tilde{\boldsymbol{\Sigma}}) \circ \tilde{\boldsymbol{\Sigma}} \circ \mathrm{diag}(\tilde{\boldsymbol{\Sigma}})^{\top}\right] \circ \boldsymbol{A}_3^{\top}
\end{aligned}$$

*Proof.* The expected values can be solved using central moments of Gaussian distributions and Isserlis' theorem. All power operations are element-wise. Note that since $\boldsymbol{A}_s$ are $n$ dimensional vector, and all the power and product operations are element-wise, the complexity of calculating covariance is $\mathcal{O}(n^2)$. For detailed derivation, see Appendix A.5.2. □

It is useful to note that an addition (residual or recurrent) layer sums the activation of two (or more) layers, e.g. $\mathbf{X}$ and $g(\mathbf{Y})$. In thise case, the covariance of $\mathbf{X} + g(\mathbf{Y})$ is the sum of their covariances and cross-covariances, i.e.

$$\boldsymbol{\Sigma}_{\mathbf{X}+g(\mathbf{Y})} = \boldsymbol{\Sigma}_{\mathbf{X}} + \boldsymbol{\Sigma}_{g(\mathbf{Y})} + \boldsymbol{\Sigma}_{\mathbf{X}g(\mathbf{Y})} + \boldsymbol{\Sigma}_{g(\mathbf{Y})\mathbf{X}}$$

It it thus helpful to postulate an additional lemma for the purpose of calculating covariance after addition.

**Lemma 2.** *Let $\mathbf{Y} \coloneqq (\mathrm{Y}_1, \mathrm{Y}_2, \cdots, \mathrm{Y}_n)^{\top}$ be another Gaussian random vector that is cross-correlated to $\mathbf{X}$ with $\boldsymbol{\Sigma}_{\mathbf{YX}}$, and $\boldsymbol{\Omega} \coloneqq \mathbf{Y} - \mathbb{E}[\mathbf{Y}]$. Then, the cross-covariance matrix between $\mathbf{Y}$ and $\mathbf{Z} \coloneqq g(\mathbf{X})$ is*

$$\begin{aligned}
\boldsymbol{\Sigma}_{\mathbf{YZ}} &= \sum_{t=1, t\ is\ odd}^{S} \boldsymbol{A}_t^{\top} \circ \mathbb{E}\left[\boldsymbol{\Omega}\ \boldsymbol{\Xi}_{(t)}^{\circ t\top}\right] \\
\boldsymbol{\Sigma}_{\mathbf{ZY}} &= \sum_{s=1, s\ is\ odd}^{S} \boldsymbol{A}_s \circ \mathbb{E}\left[\boldsymbol{\Xi}_{(s)}^{\circ s}\ \boldsymbol{\Omega}^{\top}\right]
\end{aligned} \tag{3}$$

*for an $S$-th order expansion. For $S = 3$,*

$$\begin{aligned}
\boldsymbol{\Sigma}_{\mathbf{YZ}} &= \boldsymbol{A}_1^{\top} \circ \boldsymbol{\Sigma}_{\mathbf{YX}} + 3\boldsymbol{A}_3^{\top} \circ \boldsymbol{\Sigma}_{\mathbf{YX}} \circ \mathrm{diag}(\boldsymbol{\Sigma}_{\mathbf{X}})^{\top} \\
\boldsymbol{\Sigma}_{\mathbf{ZY}} &= \boldsymbol{A}_1 \circ \boldsymbol{\Sigma}_{\mathbf{XY}} + 3\boldsymbol{A}_3 \circ \boldsymbol{\Sigma}_{\mathbf{XY}} \circ \mathrm{diag}(\boldsymbol{\Sigma}_{\mathbf{X}})
\end{aligned}$$

*Proof.* The expected value can be calculated using Isserlis' theorem. Note that this term is nonzero only if $t$ and $s$ are odd. For details of derivation, see Appendix A.5.3. □

With these results, to find the covariance of the output of a nonlinear layer, assuming the input follows a multi-variate normal distribution, one needs to derive the coefficients of the PTPE, i.e., $\boldsymbol{A}_0$, $\boldsymbol{A}_1$, $\boldsymbol{A}_2$, etc., for the nonlinearity of interest. Note that these coefficients only depend on mean $\tilde{\boldsymbol{\mu}}$ and variance $\tilde{\boldsymbol{\sigma}}^2$, not correlations, rendering the computational complexity $\mathcal{O}(n)$. We briefly discuss some of the techniques we adopted to solve for these polynomial coefficients and list the final results in Table 1 and Table 2. For detailed derivation for all nonlinearities, see Appendix A.6 - A.11.

Table 1: First four coefficients of the polynomials for seven commonly used nonlinearities. For notational simplicity, all the product, division, and power operations are element-wise.

| | $A_0$ | $A_1$ | $A_2$ | $A_3$ | $\hat{\sigma}_j^2 = \tilde{\sigma}^2 + \acute{\sigma}_j^2$ |
|---|---|---|---|---|---|
| Tanh | $\frac{1}{p}\sum_{j=1}^p (2C_j - 1)$ | $\frac{1}{p}\sum_{j=1}^p 2B_j$ | $\frac{1}{2}\frac{1}{p}\sum_{j=1}^p -2B_j\frac{\tilde{\mu}}{\hat{\sigma}_j^2}$ | $\frac{1}{3!}\frac{1}{p}\sum_{j=1}^p 2B_j\frac{\tilde{\mu}^2 - \hat{\sigma}_j^2}{\hat{\sigma}_j^4}$ | $\acute{\sigma}_j^2 = \frac{1}{2\gamma_j^2}$ |
| Sigmoid | $\frac{1}{p}\sum_{j=1}^p C_j$ | $\frac{1}{p}\sum_{j=1}^p B_j$ | $\frac{1}{2}\frac{1}{p}\sum_{j=1}^p -B_j\frac{\tilde{\mu}}{\hat{\sigma}_j^2}$ | $\frac{1}{3!}\frac{1}{p}\sum_{j=1}^p B_j\frac{\tilde{\mu}^2 - \hat{\sigma}_j^2}{\hat{\sigma}_j^4}$ | $\acute{\sigma}_j^2 = \frac{1}{2\gamma_j^2}$ |
| Softplus | $\frac{1}{p}\sum_{j=1}^p C_j\tilde{\mu} + B_j\hat{\sigma}_j^2$ | $\frac{1}{p}\sum_{j=1}^p C_j$ | $\frac{1}{2}\frac{1}{p}\sum_{j=1}^p B_j$ | $\frac{1}{3!}\frac{1}{p}\sum_{j=1}^p -B_j\frac{\tilde{\mu}}{\hat{\sigma}_j^2}$ | $\acute{\sigma}_j^2 = \frac{1}{2\gamma_j^2\beta^2}$ |
| ReLU | $C\tilde{\mu} + B\hat{\sigma}^2$ | $C$ | $\frac{1}{2}B$ | $\frac{1}{3!}\left(-B\frac{\tilde{\mu}}{\hat{\sigma}^2}\right)$ | $\acute{\sigma}^2 = 0$ |
| LeakyReLU ($\theta$) | $\theta\tilde{\mu} + (1-\theta)\left(C\tilde{\mu} + B\hat{\sigma}^2\right)$ | $\theta + (1-\theta)C$ | $\frac{1-\theta}{2}B$ | $\frac{1-\theta}{3!}\left(-B\frac{\tilde{\mu}}{\hat{\sigma}^2}\right)$ | $\acute{\sigma}^2 = 0$ |
| GELU | $C\tilde{\mu} + B\tilde{\sigma}^2$ | $C + B\frac{\tilde{\mu}}{\hat{\sigma}^2}$ | $\frac{1}{2}B\left(1 + \frac{1}{\hat{\sigma}^2} - \frac{\tilde{\mu}^2}{\hat{\sigma}^4}\right)$ | $\frac{1}{3!}B\left(-\frac{\tilde{\mu}}{\hat{\sigma}^2} - \frac{3\tilde{\mu}}{\hat{\sigma}^4} + \frac{\tilde{\mu}^3}{\hat{\sigma}^6}\right)$ | $\acute{\sigma}^2 = 1$ |
| SiLU | $\frac{1}{p}\sum_{j=1}^p C_j\tilde{\mu} + B_j\tilde{\sigma}^2$ | $\frac{1}{p}\sum_{j=1}^p C_j + B_j\frac{\tilde{\mu}\acute{\sigma}_j^2}{\hat{\sigma}_j^2}$ | $\frac{1}{2}\frac{1}{p}\sum_{j=1}^p B_j\left(1 + \frac{\acute{\sigma}_j^2}{\hat{\sigma}_j^2} + \frac{\tilde{\mu}^2\acute{\sigma}_j^2}{\hat{\sigma}_j^4}\right)$ | $\frac{1}{3!}\frac{1}{p}\sum_{j=1}^p B_j\left(-\frac{\tilde{\mu}}{\hat{\sigma}_j^2} - \frac{3\tilde{\mu}\acute{\sigma}_j^2}{\hat{\sigma}_j^4} + \frac{\tilde{\mu}^3\acute{\sigma}_j^2}{\hat{\sigma}_j^6}\right)$ | $\acute{\sigma}_j^2 = \frac{1}{2\gamma_j^2}$ |

where $B_j = \frac{1}{\hat{\sigma}_j}\varphi\left(\frac{\tilde{\mu}}{\hat{\sigma}_j}\right)$ and $C_j = \frac{1}{2}\text{erfc}\left(-\frac{\tilde{\mu}}{\sqrt{2\hat{\sigma}_j^2}}\right)$ or $\Phi\left(\frac{\tilde{\mu}}{\hat{\sigma}_j}\right)$

Table 2: General solutions for the pseudo-Taylor coefficients using Hermite polynomial. For notational simplicity, all the product, division, and power operations are element-wise. The definitions of $\boldsymbol{B}_j$ and $\hat{\boldsymbol{\sigma}}_j^2$ are the same as those in Table 1.

| | |
|---|---|
| Tanh | $\boldsymbol{A}_s(s \geq 1) = \dfrac{1}{s!}\dfrac{1}{p}\sum_{j=1}^{p} 2\boldsymbol{B}_j \boldsymbol{D}_j^{(s-1)}$ |
| Sigmoid | $\boldsymbol{A}_s(s \geq 1) = \dfrac{1}{s!}\dfrac{1}{p}\sum_{j=1}^{p} \boldsymbol{B}_j \boldsymbol{D}_j^{(s-1)}$ |
| Softplus | $\boldsymbol{A}_s(s \geq 2) = \dfrac{1}{s!}\dfrac{1}{p}\sum_{j=1}^{p} \boldsymbol{B}_j \boldsymbol{D}_j^{(s-2)}$ |
| ReLU | $\boldsymbol{A}_s(s \geq 2) = \dfrac{1}{s!}\boldsymbol{B}\boldsymbol{D}^{(s-2)}$ |
| LeakyReLU$(\theta)$ | $\boldsymbol{A}_s(s \geq 2) = \dfrac{1}{s!}(1-\theta)\boldsymbol{B}\boldsymbol{D}^{(s-2)}$ |
| GELU | $\boldsymbol{A}_s(s \geq 2) = \dfrac{1}{s!}\boldsymbol{B}\left[\boldsymbol{D}^{(s-2)} - \boldsymbol{D}^{(s)}\right]$ |
| SiLU | $\boldsymbol{A}_s(s \geq 2) = \dfrac{1}{s!}\dfrac{1}{p}\sum_{j=1}^{p}\boldsymbol{B}_j\left[\boldsymbol{D}_j^{(s-2)} - \boldsymbol{D}_j^{(s)}\right]$ |

$$\text{where } \boldsymbol{D}_j^{(s)} := \left(\frac{-1}{\sqrt{2\hat{\boldsymbol{\sigma}}_j^2}}\right)^s \mathbf{H}_s\left(\frac{\tilde{\boldsymbol{\mu}}}{\sqrt{2\hat{\boldsymbol{\sigma}}_j^2}}\right)$$

**Tanh, Sigmoid, and Softplus.** Because the integral $\int \nabla\mathbf{tanh}(x)p(x)dx$ is not tractable analytically, so we make a further approximation by substituting **tanh** with the error function which is very similar but more tractable. Specifically, we propose

$$\mathbf{tanh}(x) \approx \frac{1}{p}\sum_{j=1}^{p}\mathbf{erf}[\gamma_j x]$$

where $\{\gamma_1, \cdots, \gamma_p\}$ is a set of scaling factors obtained by numerical optimization (see Eq.6 in Appendix), and the error function is defined as

$$\mathbf{erf}(x) = \frac{2}{\sqrt{\pi}}\int_0^x \exp(-t^2)dt$$

Then, the integral $\int \nabla\mathbf{tanh}(x)p(x)dx$ can be approximated as $\frac{1}{p}\sum_{j=1}^{p}\int\nabla\mathbf{erf}(\gamma_j x)p(x)dx$, which is tractable analytically. The higher order derivatives of the error function are simply derivatives of Gaussian functions $\varphi(x)$, which are related to Hermite polynomials $\mathbf{H}_s(x)$ through

$$\frac{d^s}{dx^s}\left[\frac{1}{\sigma}\varphi\left(\frac{x}{\sigma}\right)\right] = \left(\frac{-1}{\sqrt{2\sigma^2}}\right)^s \mathbf{H}_s\left(\frac{x}{\sqrt{2\sigma^2}}\right)\frac{1}{\sigma}\varphi\left(\frac{x}{\sigma}\right)$$

and

$$\mathbf{H}_0(x) = 1$$
$$\mathbf{H}_1(x) = 2x$$
$$\mathbf{H}_2(x) = 4x^2 - 2$$
$$\dots$$

and we showed the pseudo-Taylor coefficients are convolutions of Gaussian derivatives and Gaussian pdf, which are analytically tractable A.6. We used a similar treatment for **sigmoid** function (see Appendix A.7). Using error functions as a approximation was first suggested by Spiegelhalter and Lauritzen (1990), but we use a linear combination, which is easily parallelizable and enhanced approximation accuracy. The derivation for **softplus** can reuse the results of **sigmoid**, because the derivative of **softplus** is just **softplus** with a scaling factor $\beta$ (A.8).

**ReLU and LeakyReLU.** It is obvious that we cannot apply our method directly on **ReLU**, because it is not continuously differentiable. Hence, we modified the results for **softplus** at the limit of $\beta \to \infty$, considering the relationship (A.9)

$$\lim_{\beta \to \infty} \frac{1}{\beta} \mathbf{log}\left(1 + e^{\beta x}\right) = \max\{0, x\}$$

Similarly, leaky ReLU and any piece-wise linear activation function can be described as a combination of ReLU functions with different scaling, shifting, and/or mirroring.

**GELU and SiLU.** The derivatives of GELU function can be expressed using derivatives of Gaussian cdf $\boldsymbol{\Phi}(x)$

$$\frac{\partial^s}{\partial x^s} \mathbf{GELU}(x) = s \frac{\partial^{s-1}}{\partial x^{s-1}} \boldsymbol{\Phi}(x) + x \frac{\partial^s}{\partial x^s} \boldsymbol{\Phi}(x)$$

The expected value of the GELU derivative involves integrating the product of Hermite polynomials of $x$ and Gaussian functions, which is analytically tractable (A.10). Using normal cumulative density functions to approximate a sigmoid function, the derivations for a SiLU function becomes similar to that of the GELU (A.11).

## 3 Results

### 3.1 PTPE significantly improves estimation accuracy when exposed to higher input variance

As an initial empirical test and demonstration of concept, we applied PTPE to a single, univariate nonlinearity subject to a parameterized normally distributed input. We varied the input mean and variance and examined how the output mean and variance compared to those predicted by PTPE. For this comparison, the true output statistics were obtained through $10^7$ Monte Carlo sampling across all input parameters. As expected, PTPE far outstrips Jacobian linearization, and this effect is prominent especially when input variance is high. With up-to third order PTPE, the estimated variance by our method is already very close to the ground truth (Fig. 3 col 4).

### 3.2 PTPE accurately quantifies uncertainty in canonical network architectures

To benchmark PTPE for uncertainty quantification in neural networks, we trained 9 residual neural networks (He et al. (2016)) with three depths (13, 33, and 65 layers) and 3 three typical nonlinearities (ReLU, GELU, Tanh) on CIFAR10 (Krizhevsky (2009)). We corrupted each input image with additive Gaussian noise, then compared the PTPE-predicted and true (via Monte Carlo sampling) output distributions (Fig. 4). The layerwise application of PTPE is outline in Algorithm 1 with accompanying pseudo code.

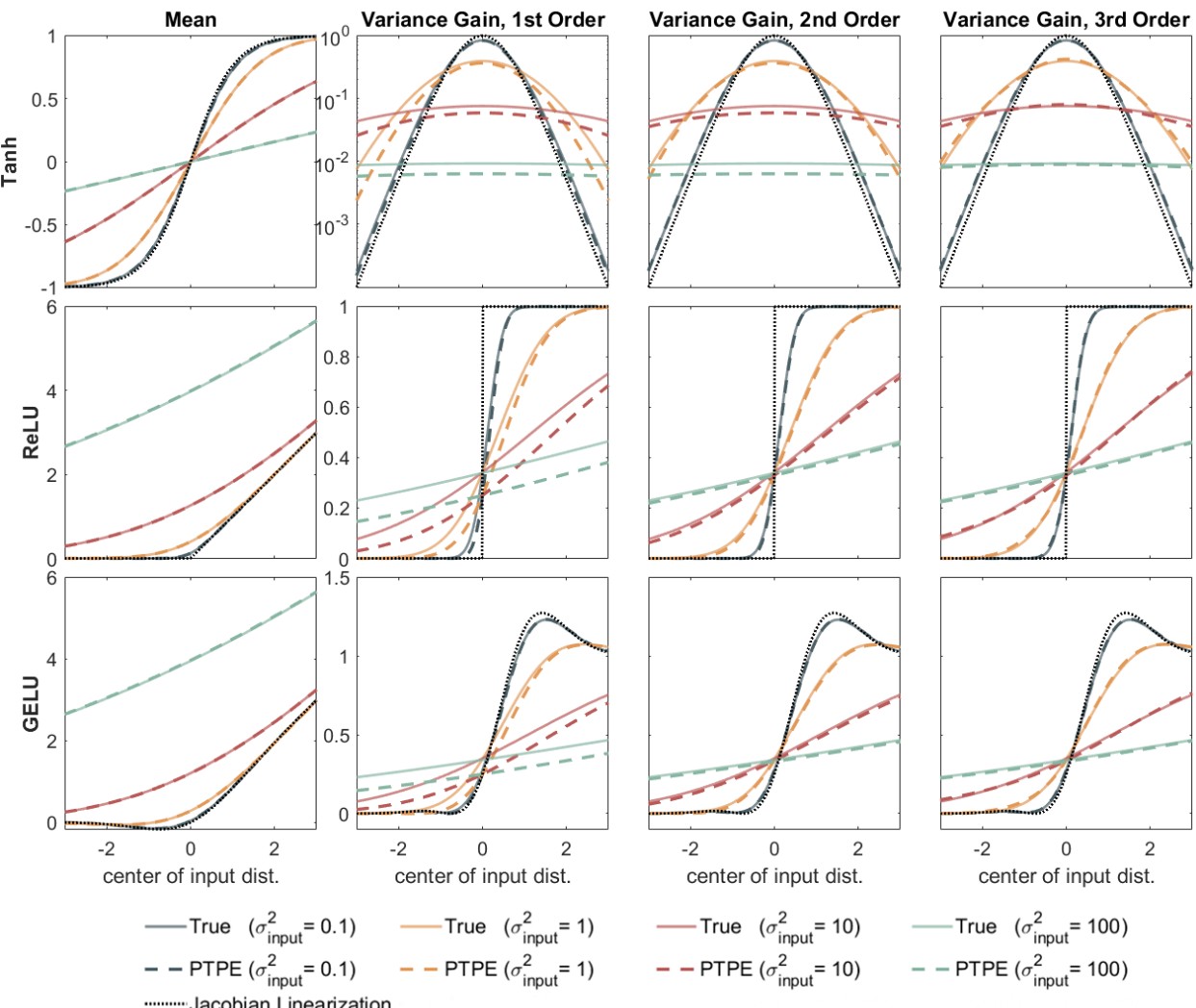

Figure 3: **Solid lines**: mean (column 1) and variance gain (output variance divided by input variance) (column 2-5) obtained by sampling 1e6 datapoints from Gaussian distributions with centers ranging from -3 to 3. **Dashed lines**: approximated mean and variance gain predicted with 1st, 2nd, and 3rd order pseudo Taylor polynomial expansion (column 2 - 4). Colors correspond to different input variances (blue: 0.1, yellow: 1, red: 10, green: 100). **Dotted lines**: approximated mean and variance gain using Jacobian linearization (first order deterministic Taylor expansion around input mean)

We measure the accuracy of the output covariance matrix in two ways. First, we evaluate the Frobenius norm of the residuals $||\mathbf{\Sigma}_{\text{est}} - \mathbf{\Sigma}_{\text{true}}||_{\text{fro}}$. Then, we fit Gaussian models with the (estimated or true) mean and covariance, and examine the Kullback-Leibler divergence between the two distributions, as a holistic measurement. We summarize the results in Table 3.

Overall, the experimental results align with expectations: (1) Jacobian linearization degrades dramatically in moderate to high variance regime. (2) Direct derivation is not suitable for this task due to the assumption of independence, since the overlapping convolution kernels and residual layers introduce substantial correlation. (3) Introducing up-to the third order PTPE typically outperformes stochastic and Jacobian linearization by a large margin. In the specific case of resnet13 with ReLU nonlinearity, stochastic linearization surpasses

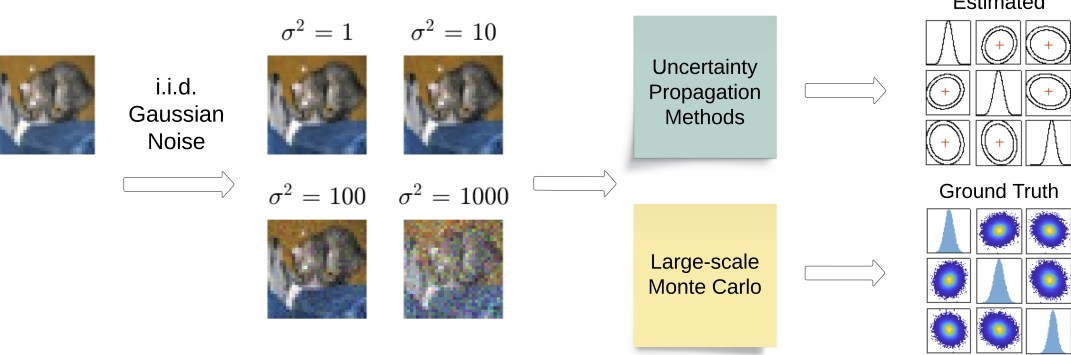

Figure 4: Schematic of experiment setup. We inject i.i.d. Gaussian noise to input image to simulate sensor noises, then compare the estimated output distribution to the ground truth obtained by large-scale simulation (sampling $10^7$ noisy images).

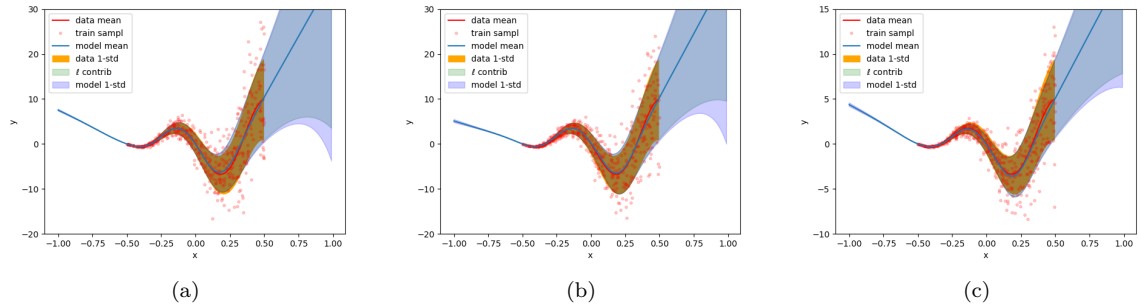

Figure 5: We performed the 1D regression task by training neural networks using DVI with three different nonlinear activations. The estimation of the posterior mean and covariance is implemented with up to third order of PTPE. We show the fitting obtained by neural networks with (a) Tanh (b) ReLU (b) GELU, respectively.

PTPE by a small margin. We surveyed the input variance prior to each nonlinear layer and found it well below 0.1 in almost all cases, i.e., this turns out to be a low-noise model.

### 3.3 PTPE addresses the limitations of DVI by incorporating non-piecewise-linear activations.

One major contribution of our work is to address the lack of accurate deterministic moment estimation for general nonlinearities in the field of variational inference. The method suggested by Hernández-Lobato and Adams (2015) Wu et al. (2019) share the same goal, which is to find a deterministic method to approximate moments in neural networks. However, closed form solutions of posterior mean and covariance were solved only for piecewise linear activations such as ReLU and Heaviside. We know $\mathrm{Var}(f(X)) = \mathbb{E}[f(X)^2] - \mathbb{E}[f(X)]^2$, and the first term, $\mathbb{E}[f(X)^2] = \int f(x)^2 p_X(x)dx$, becomes arduous to solve for more complex nonlinearities $f(\cdot)$. Our approach circumvents this issue by taking derivatives inside the integrals, which provides tractability for general nonlinearities.

We provide additional context and quantification, by integrating PTPE into deterministic variational inference (DVI) Wu et al. (2019) and conduct 1D regression experiments (Figure 5). This regression task is the same to that of Wu et al. (2019) except the data variance is higher. The strong alignment between true and predicted intervals demonstrates PTPE's effectiveness and reliability for variational inference applications.

Table 3: Estimation accuracies of uncertainty propagation methods evaluated on residual networks of three different depth, three different nonlinearities, and four different input variance. The smaller the better. Direct derivation of GELU is not reported because it is not found in literature.

| | | | Stochastic or Deterministic Expansion | | | | | | | | Direct Derivation (assume independence) | |
| | | | 3rd order PTPE (Ours) | | 2nd order PTPE (Ours) | | 1st order PTPE (SL) e.g. Zhen et al. (2021) | | Deterministic 1st order Taylor e.g. Petersen et al. (2024) | | ReLU Frey and Hinton (1999) Tanh Wang et al. (2016) | |
| | Nonlin. | Input Var. | KLdiv | Frob. Cov | KLdiv | Frob. Cov | KLdiv | Frob. Cov | KLdiv | Frob. Cov | KLdiv | Frob. Cov |
|---|---|---|---|---|---|---|---|---|---|---|---|---|
| ResNet13 | ReLU | 1 | 1.9830 | 0.0252 | 1.9742 | 0.0227 | **1.9555** | **0.0130** | 12956 | 1686.1 | Inf | 0.2636 |
| | | 10 | 6.8032 | 0.4116 | 6.7333 | 0.3716 | **6.6570** | **0.1811** | 18632 | 16861 | 38.230 | 1.8590 |
| | | 100 | 9.2222 | 3.3886 | 9.0892 | 2.8985 | **8.8476** | **1.5071** | 57634 | 1.6862e5 | 53.536 | 6.3657 |
| | | 1000 | 75.545 | 8.7788 | 60.770 | 7.3594 | **46.688** | **3.4739** | -Inf | 1.6863e6 | 145.13 | 6.4964 |
| | GELU | 1 | 0.1641 | 0.0567 | 0.0734 | 0.0371 | **0.0427** | **0.0254** | 0.7556 | 0.0597 | N/A | |
| | | 10 | 0.2606 | 0.4509 | 0.0765 | 0.1867 | **0.0524** | **0.1212** | 5.7486 | 1.2599 | | |
| | | 100 | 0.7204 | 1.3371 | **0.2109** | **0.6074** | 1.2855 | 2.1094 | 44.531 | 23.552 | | |
| | | 1000 | **3.3989** | **1.1458** | 4.1200 | 2.0713 | 9.6970 | 3.6949 | 516.30 | 284.47 | | |
| | Tanh | 1 | **0.0398** | 0.0119 | 0.0412 | **0.0114** | 0.0538 | 0.0143 | 0.7343 | 0.0455 | 91.409 | 0.1259 |
| | | 10 | **0.2970** | **0.0698** | 0.3294 | 0.0953 | 0.4574 | 0.3265 | 5.2699 | 1.2315 | 13.216 | 1.2397 |
| | | 100 | **1.1637** | **0.3522** | 1.3214 | 0.6513 | 2.1006 | 1.9951 | 32.560 | 21.456 | 17.948 | 4.1842 |
| | | 1000 | 2.6870 | **0.8039** | **2.5230** | 1.4988 | 8.2010 | 3.6733 | 260.97 | 251.06 | 55.604 | 5.0782 |
| ResNet33 | ReLU | 1 | 1.7206 | **0.0135** | 1.7135 | 0.0136 | **1.2553** | 0.0173 | -Inf | 3.9662e5 | Inf | 0.1910 |
| | | 10 | 5.9533 | **0.1947** | 5.9299 | 0.2051 | **5.0286** | 0.3522 | -Inf | 3.9662e6 | Inf | 1.4182 |
| | | 100 | 8.0600 | **2.2561** | 8.2479 | 2.4993 | **6.8999** | 3.5706 | -Inf | 3.9662e7 | Inf | 6.7729 |
| | | 1000 | 27.182 | **3.4422** | 20.965 | 3.5344 | **20.258** | 3.9930 | -Inf | 3.9662e8 | 187.10 | 5.6828 |
| | GELU | 1 | 0.6193 | **0.0073** | **0.3450** | 0.0197 | 0.3496 | 0.0198 | 0.5941 | 0.0189 | N/A | |
| | | 10 | **0.6316** | **0.0773** | 0.6662 | 0.2240 | 0.7479 | 0.2313 | 2.8821 | 0.1730 | | |
| | | 100 | **1.4799** | **1.2352** | 3.2127 | 2.3834 | 4.0093 | 2.5805 | 15.650 | 1.2557 | | |
| | | 1000 | **3.0118** | **1.4099** | 7.9264 | 2.1759 | 10.2863 | 2.5945 | 136.74 | 38.561 | | |
| | Tanh | 1 | 0.1738 | 0.0297 | **0.1705** | **0.0296** | 0.2053 | 0.0339 | 0.8907 | 0.0287 | 266.79 | 0.0972 |
| | | 10 | **0.5381** | 0.3440 | 0.5520 | 0.3442 | 0.8110 | 0.5163 | 7.0697 | 0.4308 | 57.076 | 1.2637 |
| | | 100 | **1.7506** | **2.6232** | 2.0170 | 2.7172 | 3.3515 | 3.8600 | 23.278 | 10.729 | 37.783 | 6.2437 |
| | | 1000 | **3.3458** | **5.1601** | 3.5783 | 5.4889 | 7.4666 | 8.1504 | 111.59 | 157.09 | 58.041 | 11.086 |
| ResNet65 | ReLU | 1 | 1.4917 | **0.0467** | 1.4814 | 0.0479 | **0.8827** | 0.0432 | -Inf | 1.4451e6 | Inf | 0.2658 |
| | | 10 | 8.4157 | **0.6900** | 8.2384 | 0.7003 | **6.6310** | 0.8260 | -Inf | 1.4451e7 | Inf | 2.0032 |
| | | 100 | **9.2703** | **4.7281** | 9.9622 | 4.8751 | 10.768 | 5.5521 | -Inf | 1.4451e8 | Inf | 9.0630 |
| | | 1000 | **24.553** | **6.9253** | 26.293 | 6.9553 | 43.585 | 7.1456 | -Inf | 1.4451e9 | 149.03 | 8.4401 |
| | GELU | 1 | **0.5624** | **0.0487** | 0.7439 | 0.0587 | 0.8083 | 0.0601 | 23812 | 0.1246 | N/A | |
| | | 10 | **1.3514** | **0.4572** | 1.4035 | 0.5509 | 1.6731 | 0.5903 | 2222.1 | 0.9907 | | |
| | | 100 | **2.4530** | **1.7822** | 2.6217 | 2.3779 | 3.7498 | 2.8196 | 19.722 | 4.8688 | | |
| | | 1000 | **4.5847** | **1.8153** | 7.2416 | 2.6670 | 11.949 | 3.2836 | 201.36 | 84.473 | | |
| | Tanh | 1 | **1.2616** | 0.1042 | 1.2644 | 0.1041 | 1.3358 | 0.1073 | 1.5176 | **0.0872** | 693.30 | 0.1426 |
| | | 10 | **1.8372** | 0.6403 | 1.8704 | 0.6431 | 2.4041 | 0.7024 | 5.7769 | **0.4136** | 113.62 | 1.0517 |
| | | 100 | **3.9667** | **2.2011** | 4.2400 | 2.2295 | 5.4892 | 2.4843 | 33.5632 | 10.024 | 79.875 | 3.3743 |
| | | 1000 | **4.9223** | **3.2582** | 5.1105 | 3.3049 | 7.7989 | 3.7953 | 168.3758 | 124.67 | 122.56 | 4.7517 |

## 4 Discussion

One immediate potential limitation of PTPE is its reliance on the assumption that inputs are Gaussian. It has been well-established that at the limit of infinite width, a deep neural network with Gaussian input is equivalently a Gaussian process (Neal (1994); Williams (1997); de G. Matthews et al. (2018); Lee et al. (2018); Gao et al. (2023)), and a similar phenomenon is also reported in Bayesian neural networks with Gaussian weights (Goulet et al. (2021); Nguyen and Goulet (2022)). Based on this observation, we assume a "wide enough" neural network will have approximate Gaussianity in each layer, so that the error of using variational Gaussian distributions to approximate layer-wise distributions becomes negligible. We verify this assumption through simulation (see e.g., Fig. 6).

In this paper, we focus on the propagation of Gaussian distributions. This choice is due to their prevalence in machine learning and their convenient property of being Lévy alpha-stable, meaning a linear combination of Gaussian random variables remains Gaussian. This makes Gaussian distributions pertinent to our objectives. Consequently, our method could potentially be extended to other types within the Lévy alpha-stable family. For instance, Peterson et. al. demonstrated the propagation of Cauchy distributions through neural networksPetersen et al. (2024). A more comprehensive survey is provided in Wang et al. (2016), where the authors examined the propagation of exponential family distributions (including Beta, Rayleigh, Gamma, Poisson, and Gaussian), though this requires more intricate derivations.

A strength of PTPE is its generality. As mentioned in the introduction, several immediate motivating use-cases are in the training of robust networks including probabilistic network models. Furthermore, our proposed method may also find application in safety-critical engineering systems

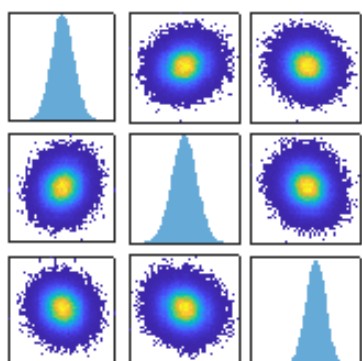

Figure 6: Empirical distributions (of the first three units) before the final softmax layer of a residual neural net trained on CIFAR10. This network has 13 ReLU layers. The distributions are obtained by Monte Carlo sampling $10^5$ images with additive Gaussian noises. For distributions of all 10 units, see Fig. 7 in Appendix.

that require estimates on uncertainty. Recently, researchers combined an LSTM and Kalman filtering to monitor the states of plasma inside a nuclear fusion device Pavone et al. (2023). The Kalman filter, by construction, requires statistics on the output of the LSTM in order to generate control signals. Such statistics were generated by using a probabalistic architcture within the LSTM, i.e., where parameters are specified by a learned distribution. PTPE provides a potential alternative path for such problems, but enabling uncertainty propagation through deterministic learned architectures.

## 5 Conclusion

In this article, we developed a stochastic polynomial expansion approach, PTPE, to perform uncertainty propagation in neural networks. Our method offers significant advantages in accurately propagating the full covariance matrix of an input distribution compared to state-of-the-art methods, without substantially sacrificing computational efficiency. We derived analytical solutions for the first two moments of the output distributions for seven commonly used nonlinearities, demonstrating remarkable accuracy in predicting univariate mean and variance, particularly under high uncertainty. Additionally, we assessed its multivariate accuracy in deep residual neural networks trained on image categorization tasks. By incorporating up to third-order polynomial expansion, our method generally outperformed others, except in scenarios with minimal uncertainty in which the performance of competing methods is comparable. Overall, our proposed method provides a tractable framework for solving uncertainty propagation problems. It can potentially be effectively applied in various domains, including adversarial training, Bayesian inference, and safety-critical applications, offering a versatile tool for enhancing the reliability and robustness of neural networks.

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

# A   Appendix

## A.1   Emperical distributions of ResNets show Gaussianity

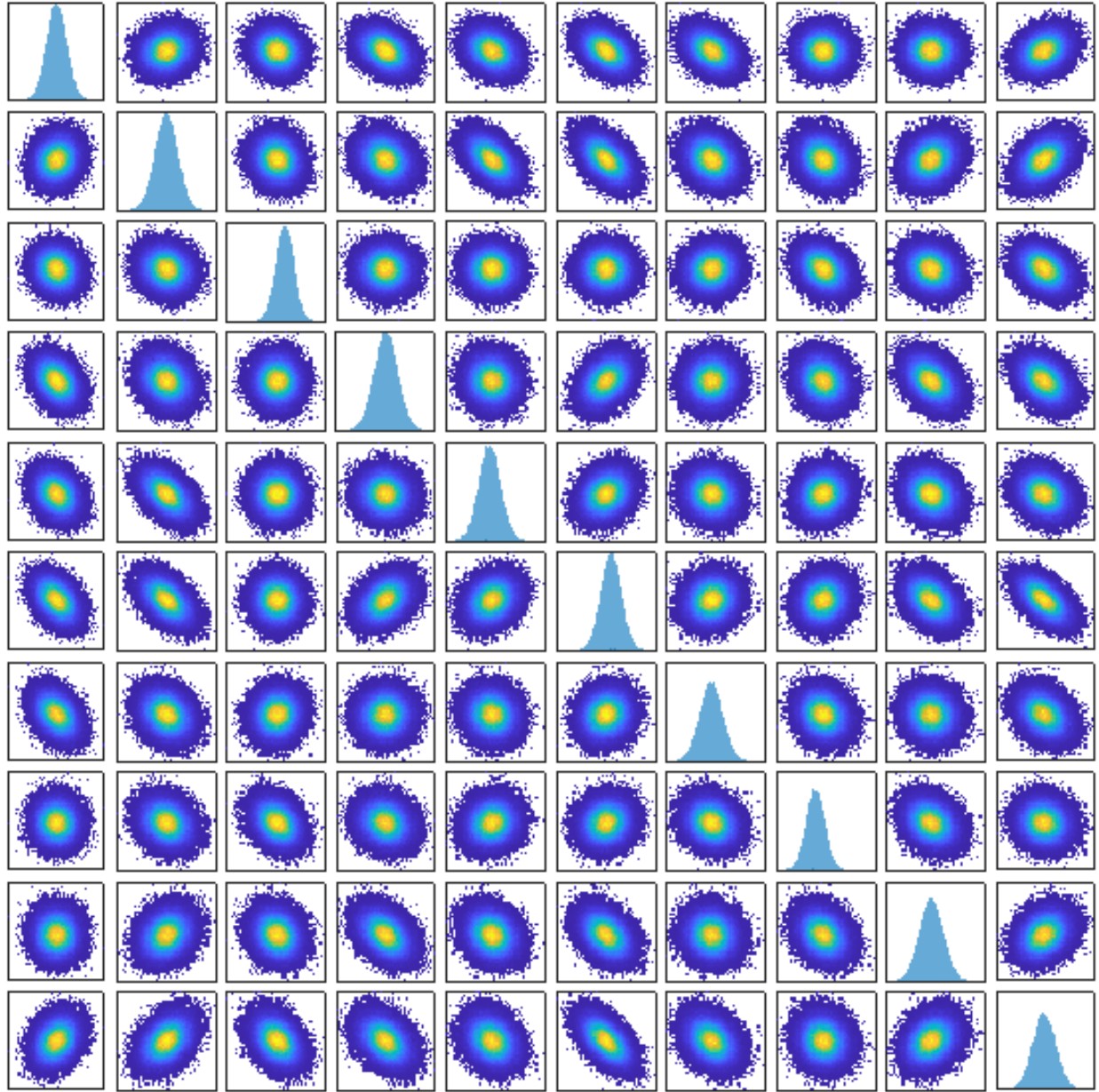

Figure 7: Empirical distributions of all units before the final softmax layer of the resnet13(ReLU).

## A.2   Approximation accuracy on other non-linearity

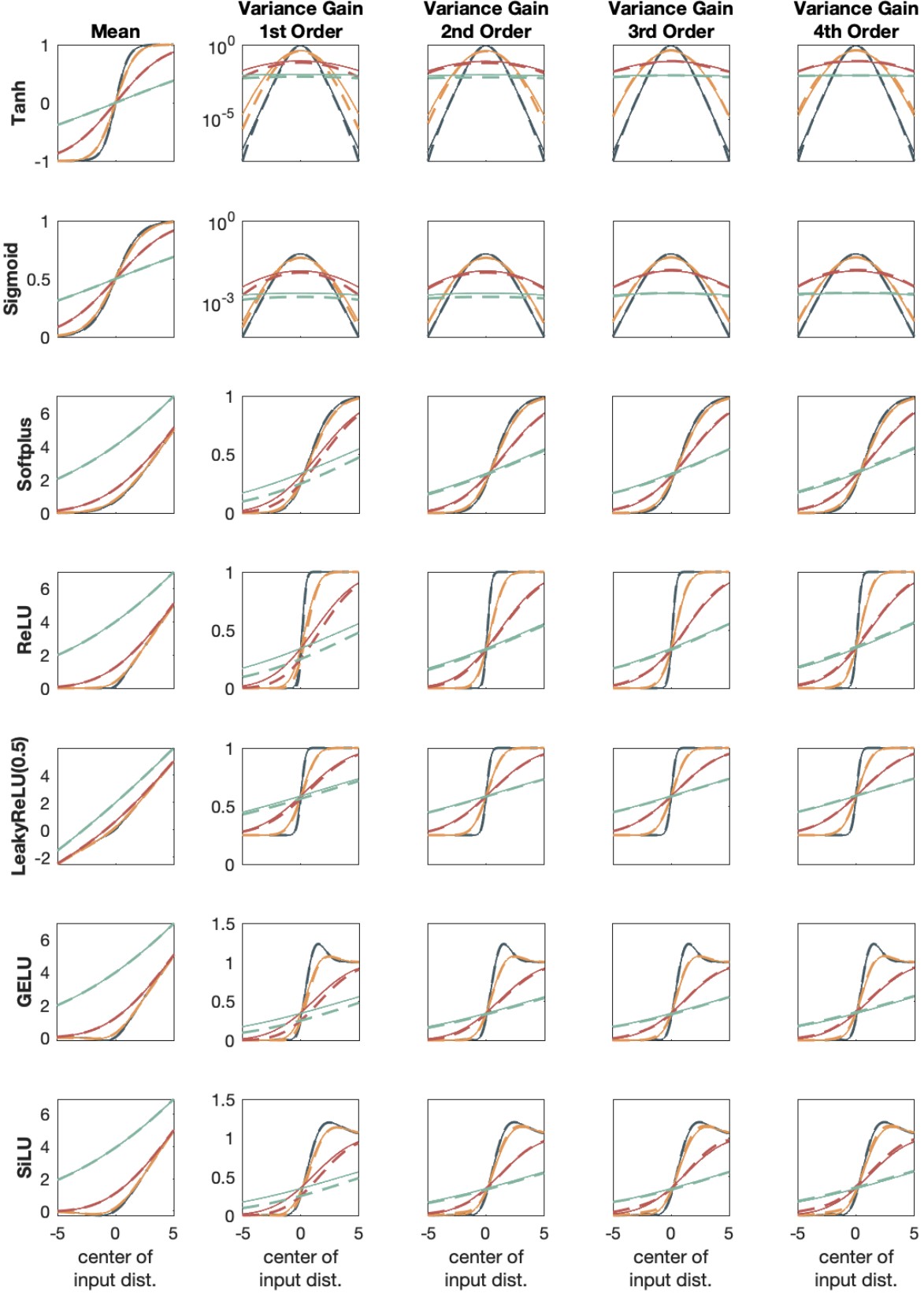

Figure 8: The meaning of different colors and line styles is the same as Fig. 3. The higher the input variance, the more significant is the benefit of using higher order Stochastic Taylor expansion.

## A.3 Pseudo Code

---

**Algorithm 1** Propagating a multi-variate Gaussian distribution through a pretrained ResNet

---

$\mu \leftarrow$ Input mean
$\Sigma \leftarrow$ Input covariance
$\mu_{res} \leftarrow$ Storage for mean of the output of residual layer
$\Sigma_{res} \leftarrow$ Storage for covariance of the output of residual layer
$\Sigma_{cross} \leftarrow$ Storage for cross-covariance between the two input of the residual layer
**for** layer in neural network **do**
  **if** layer is linear **then**
    **if** layer is addition (residual) **then**
      $\mu \leftarrow \mu + \mu_{res}$
      $\Sigma \leftarrow \Sigma + \Sigma_{res} + \Sigma_{cross} + \Sigma_{cross}^{\top}$
      empty $\mu_{res}, \Sigma_{res}, \Sigma_{cross}$
    **else**
      find effective weight $W$ and bias $b$
      $\mu \leftarrow W^{\top}\mu + b$
      $\Sigma \leftarrow W^{\top}\Sigma W$
      $\Sigma_{cross} \leftarrow W^{\top}\Sigma_{cross}$
      **if** residual connection starts from here **then**
        $\mu_{res} \leftarrow \mu$
        $\Sigma_{res}, \Sigma_{cross} \leftarrow \Sigma$
      **end if**
    **end if**
  **else**
    $\mu, \Sigma, \Sigma_{cross} \leftarrow \mathrm{PTPE}(\text{nonlinearity}, \mu, \Sigma, \Sigma_{cross})$
    **if** residual connection starts from here **then**
      $\mu_{res} \leftarrow \mu$
      $\Sigma_{res}, \Sigma_{cross} \leftarrow \Sigma$
    **end if**
  **end if**
**end for**
$\mu_{output} \leftarrow \mu$
$\Sigma_{output} \leftarrow \Sigma$

---

### A.4 Notations

Revisit the meaning of notations.

$$
\begin{array}{rl}
\mathbf{U} & \text{multivariate Gaussian input} \quad \sim \mathcal{N}^n(\boldsymbol{\mu}, \boldsymbol{\Sigma}) \\
\boldsymbol{W} & \text{weight matrix (constant)} \\
\boldsymbol{b} & \text{bias vector (constant)} \\
\mathbf{X} & = \boldsymbol{W}^\top \mathbf{U} + \boldsymbol{b} \quad \sim \mathcal{N}^n(\tilde{\boldsymbol{\mu}}, \tilde{\boldsymbol{\Sigma}}) \\
\tilde{\boldsymbol{\mu}} & = \boldsymbol{W}^\top \boldsymbol{\mu} + \boldsymbol{b} \\
\tilde{\boldsymbol{\Sigma}} & = \boldsymbol{W}^\top \boldsymbol{\Sigma} \boldsymbol{W} \\
\boldsymbol{\Xi} & = \mathbf{X} - \tilde{\boldsymbol{\mu}} \quad \sim \mathcal{N}^n(\mathbf{0}, \tilde{\boldsymbol{\Sigma}}) \\
\tilde{\boldsymbol{\sigma}}^2 & = \operatorname{diag}\left(\tilde{\boldsymbol{\Sigma}}\right) \\
\{\gamma_1, \dots, \gamma_p\} & \text{positive scaling factor set obtained through numerical optimizations}
\end{array}
$$

For notationaly simplicity, all product, division, and power operations are element-wise starting from A.6.

### A.5 Mean, Covariance, and Cross-covariance propagated through a univariate nonlinear function

In the context of machine learning, all non-linearities are applied element-wise – they are univarite. Thus, the off-diagonal entries of their Hessian matrices (second order partial derivatives) are zero, and it has similar effect on higher order partial derivatives. This makes PTPE on multivariate input easier to write down.

Given a smooth nonlinear function $\bar{f}(\cdot)$ of univariate random variables, define vector operation $f(\mathbf{X}) :=(\bar{f}(\mathrm{X}_1), \bar{f}(\mathrm{X}_2), \cdots, \bar{f}(\mathrm{X}_n))^\top$. We define an approximation $g(\cdot)$, which stochastically expands $f(\cdot)$ under the Taylor scheme, such that $f(\cdot)$ and $g(\cdot)$ have approximately the same first and second moment. This expansion uses i.i.d. surrogate polynomials, $\{\mathbf{1}, \boldsymbol{\Xi}_{(1)}, \boldsymbol{\Xi}_{(2)}^{\circ 2}, \boldsymbol{\Xi}_{(3)}^{\circ 3}, \cdots\}$, and such choice reduces computational complexity of covariance, which is shown later.

$$
g\left(\mathbf{X}\right) = \mathbb{E}\left[f\left(\mathbf{X}\right)\right] + \frac{\mathbb{E}\left[\nabla_{\mathbf{x}} f\left(\mathbf{X}\right)\right]}{1!} \circ \left(\boldsymbol{\Xi}_{(1)} - \mathbb{E}\left[\boldsymbol{\Xi}_{(1)}\right]\right) + \frac{\mathbb{E}\left[\nabla_{\mathbf{x}}^2 f\left(\mathbf{X}\right)\right]}{2!} \circ \left(\boldsymbol{\Xi}_{(2)}^{\circ 2} - \mathbb{E}\left[\boldsymbol{\Xi}_{(2)}^{\circ 2}\right]\right) + \cdots
$$

and denote

$$
\begin{aligned}
\boldsymbol{A}_0 &= \mathbb{E}\left[f\left(\mathbf{X}\right)\right] \\
\boldsymbol{A}_1 &= \frac{\mathbb{E}\left[\nabla_{\mathbf{x}} f\left(\mathbf{X}\right)\right]}{1!} \\
\boldsymbol{A}_2 &= \frac{\mathbb{E}\left[\nabla_{\mathbf{x}}^2 f\left(\mathbf{X}\right)\right]}{2!} \\
\boldsymbol{A}_3 &= \frac{\mathbb{E}\left[\nabla_{\mathbf{x}}^3 f\left(\mathbf{X}\right)\right]}{3!} \\
&\cdots
\end{aligned}
$$

such that

$$
g\left(\mathbf{X}\right) = \mathbb{E}\left[f\left(\mathbf{X}\right)\right] + \underbrace{\sum_{s=1}^{\infty} \boldsymbol{A}_s \circ \left(\boldsymbol{\Xi}_{(s)}^{\circ s} - \mathbb{E}\left[\boldsymbol{\Xi}_{(s)}^{\circ s}\right]\right)}_{\text{zero mean}}
$$

#### A.5.1 Mean

$\boldsymbol{A}_0$ is simply the mean of the output. In the later sections we show this value can either be analytically solved or approximated using similarly behaving nonlinear functions.

### A.5.2 Covariance

For clarity of reading, we omit the subscript of $\xi$, but will revisit the independence of polynomial basis. The covariance function of $f(\mathbf{X})$ with S-th order expansion is

$$
\begin{aligned}
\mathrm{cov}\,(g(\mathbf{X})) &= \mathbb{E}\left[\left(\sum_{s=1}^{S}\boldsymbol{A}_s \circ \left(\boldsymbol{\Xi}_{(s)}^{\circ s} - \mathbb{E}[\boldsymbol{\Xi}_{(s)}^{\circ s}]\right)\right)\left(\sum_{t=1}^{S}\boldsymbol{A}_t \circ \left(\boldsymbol{\Xi}_{(t)}^{\circ t} - \mathbb{E}[\boldsymbol{\Xi}_{(t)}^{\circ t}]\right)\right)^{\top}\right] \\
&= \sum_{s=1}^{S}\sum_{t=1}^{S}\left(\boldsymbol{A}_s \boldsymbol{A}_t^{\top}\right)\circ \mathbb{E}\left[\left(\boldsymbol{\Xi}_{(s)}^{\circ s} - \mathbb{E}[\boldsymbol{\Xi}_{(s)}^{\circ s}]\right)\left(\boldsymbol{\Xi}_{(t)}^{\circ t} - \mathbb{E}[\boldsymbol{\Xi}_{(t)}^{\circ t}]\right)^{\top}\right] \\
&= \sum_{s=1}^{S}\sum_{t=1}^{S}\left(\boldsymbol{A}_s \boldsymbol{A}_t^{\top}\right)\circ \left(\mathbb{E}\left[\boldsymbol{\Xi}_{(s)}^{\circ s}(\boldsymbol{\Xi}_{(t)}^{\circ t})^{\top}\right] - \mathbb{E}[\boldsymbol{\Xi}_{(s)}^{\circ s}]\mathbb{E}[\boldsymbol{\Xi}_{(t)}^{\circ t}]^{\top}\right)
\end{aligned}
$$

which is an $n \times n$ matrix, and the $\{i,j\}$-th entry is

$$
\sum_{s=1}^{S}\sum_{t=1}^{S}A_{s,i}A_{t,j}\left(\mathbb{E}\left[\Xi_{(s),i}^{s}\Xi_{(t),j}^{t}\right] - \mathbb{E}\left[\Xi_{(s),i}^{s}\right]\mathbb{E}\left[\Xi_{(t),j}^{t}\right]\right)
$$

Since $\boldsymbol{\Xi}_{(1)}, \boldsymbol{\Xi}_{(2)}^{\circ 2}, \boldsymbol{\Xi}_{(3)}^{\circ 3}, \cdots$ are independent, the off-diagonal entries $(s \neq t)$ are zero, then the $\{i,j\}$-th entry becomes

$$
\sum_{s=1}^{S}A_{s,i}A_{s,j}\left(\mathbb{E}\left[\Xi_{(s),i}^{s}\Xi_{(s),j}^{s}\right] - \mathbb{E}\left[\Xi_{(s),i}^{s}\right]\mathbb{E}\left[\Xi_{(s),j}^{s}\right]\right)
$$

Rewrite in matrix form

$$
\mathrm{cov}\,(g(\mathbf{X})) = \sum_{s=1}^{S}\boldsymbol{A}_s \circ \left(\mathbb{E}\left[\boldsymbol{\Xi}_{(s)}^{\circ s}\,\boldsymbol{\Xi}_{(s)}^{\circ s\,\top}\right] - \mathbb{E}\left[\boldsymbol{\Xi}_{(s)}^{\circ s}\right]\mathbb{E}\left[\boldsymbol{\Xi}_{(s)}^{\circ s}\right]^{\top}\right)\circ \boldsymbol{A}_s^{\top} \tag{4}
$$

where $\boldsymbol{A}_s$ and $\boldsymbol{\Xi}_{(s)}$ are both n dimensional vertical vectors. Using central moments of normal distributions,

$$
\mathbb{E}\left[\Xi_{(s),i}^{s}\right] = \begin{cases} 0 & \text{if } s \text{ is odd} \\ \tilde{\sigma}_i^s(s-1)!! & \text{if } s \text{ is even} \end{cases}
$$

With application of Isserlis' theorem,

$$
\mathbb{E}\left[\Xi_{(s),i}^{s}\Xi_{(s),j}^{s}\right] = \sum_{p\in P_{\mathcal{B}}^2}\prod_{\{c,d\}\in p}\tilde{\rho}_{cd}\tilde{\sigma}_c\tilde{\sigma}_d
$$

where $c, d \in \{i, j\}$, and $\tilde{\rho}_{cd}$ is correlation. The sum is over all the pairings of the set $\mathcal{B} = \{\underbrace{i, i, \cdots, i}_{s}, \underbrace{j, j, \cdots, j}_{s}\}$, i.e. all distinct (suppose each i or j is different from other i's or j's) ways of partitioning $\mathcal{B}$ into pairs $\{c, d\}$, and the product is over the pairs contained in $p$ Janson (1997)Michalowicz et al. (2011), so there exists $(2s-1)!!$ pairs in the partition, or $(2s-1)!!$ terms in the sum. For example, the first four terms of Eqn. 4 are

$$
\begin{aligned}
&\boldsymbol{A}_1 \circ \tilde{\boldsymbol{\Sigma}} \circ \boldsymbol{A}_1^{\top} \\
&\boldsymbol{A}_2 \circ \left(2\tilde{\boldsymbol{\Sigma}}^{\circ 2}\right)\circ \boldsymbol{A}_2^{\top} \\
&\boldsymbol{A}_3 \circ \left[6\tilde{\boldsymbol{\Sigma}}^{\circ 3} + 9\,\mathrm{diag}(\tilde{\boldsymbol{\Sigma}})\circ \tilde{\boldsymbol{\Sigma}}\circ \mathrm{diag}(\tilde{\boldsymbol{\Sigma}})^{\top}\right]\circ \boldsymbol{A}_3^{\top} \\
&\boldsymbol{A}_4 \circ \left[24\tilde{\boldsymbol{\Sigma}}^{\circ 4} + 72\,\mathrm{diag}(\tilde{\boldsymbol{\Sigma}})\circ \tilde{\boldsymbol{\Sigma}}^{\circ 2}\circ \mathrm{diag}(\tilde{\boldsymbol{\Sigma}})^{\top}\right]\circ \boldsymbol{A}_4^{\top}
\end{aligned}
$$

$\boldsymbol{A}_s$ and $\mathrm{diag}(\tilde{\boldsymbol{\Sigma}})$ are both n dimensional vertical vector. With this result, to find the covariance of the output of a nonlinear layer, assuming the input follows a multi-variate normal distribution, one just needs to derive the factors of Taylor polynomial, $\boldsymbol{A}_0$, $\boldsymbol{A}_1$, $\boldsymbol{A}_2$, etc., for the nonlinearity.

### A.5.3 Cross-covariance

Let $\mathbf{Y} := (Y_1, Y_2, \cdots, Y_n)^\top$ be a Gaussian random vector that is cross-correlated to $\mathbf{X}$, and $\mathbf{\Omega} = \mathbf{Y} - \mathbb{E}[\mathbf{Y}]$. If $\mathbf{X}$ undergoes a non-linear transformation via function $f(\cdot)$,

$$\mathbf{Z} = f(\mathbf{X})$$

The cross-covariance between $\mathbf{Y}$ and $\mathbf{Z}$ can be written as

$$\mathbf{\Sigma_{YZ}} = \mathbb{E}\left[\mathbf{\Omega}\left(\sum_{t=1}^{S} \boldsymbol{A}_t \circ \left(\mathbf{\Xi}_{(t)}^{\circ t} - \mathbb{E}\left[\mathbf{\Xi}_{(t)}^{\circ t}\right]\right)\right)^\top\right]$$

which is an $n \times n$ matrix, and the $\{i, j\}$-th entry is

$$\sum_{t=1}^{S} A_{t,j}\left(\mathbb{E}\left[\Omega_i \Xi_{(t),j}^t\right] - \mathbb{E}\left[\Omega_i\right]\mathbb{E}\left[\Xi_{(t),j}^t\right]\right)$$

$\mathbb{E}[\Omega_i]$ is zero by our definition, so it can be simplified as

$$\mathbf{\Sigma_{YZ}}(i, j) = \sum_{t=1, t \text{ is odd}}^{S} A_{t,j}\mathbb{E}\left[\Omega_i \Xi_{(t),j}^t\right]$$

$$\mathbf{\Sigma_{ZY}}(i, j) = \sum_{s=1, s \text{ is odd}}^{S} A_{s,i}\mathbb{E}\left[\Xi_{(s),i}^s \Omega_j\right]$$

Rewrite in matrix form

$$\mathbf{\Sigma_{YZ}} = \sum_{t=1, t \text{ is odd}}^{S} \boldsymbol{A}_t^\top \circ \mathbb{E}\left[\mathbf{\Omega}\,\mathbf{\Xi}_{(t)}^{\circ t\,\top}\right]$$

$$\mathbf{\Sigma_{ZY}} = \sum_{s=1, s \text{ is odd}}^{S} \boldsymbol{A}_s \circ \mathbb{E}\left[\mathbf{\Xi}_{(s)}^{\circ s}\,\mathbf{\Omega}^\top\right] \tag{5}$$

and the expected value can be calculated using Isserlis' theorem mentioned above. Note that this term is nonzero only if $t$ and $s$ are odd, so the first two terms are

$$\mathbf{\Sigma_{YZ}} \approx \boldsymbol{A}_1^\top \circ \mathbf{\Sigma_{YX}} + 3\boldsymbol{A}_3^\top \circ \mathbf{\Sigma_{YX}} \circ \operatorname{diag}(\mathbf{\Sigma_X})^\top$$
$$\mathbf{\Sigma_{ZY}} \approx \boldsymbol{A}_1 \circ \mathbf{\Sigma_{XY}} + 3\boldsymbol{A}_3 \circ \mathbf{\Sigma_{XY}} \circ \operatorname{diag}(\mathbf{\Sigma_X})$$

An addition (e.g. residual) layer outputs the summation of activation of two (or more) layers, $\mathbf{Y}$ and $\mathbf{Z}$. Thus the covariance of $\mathbf{Y} + \mathbf{Z}$ is the sum of their covariance and cross-covariance.

$$\mathbf{\Sigma}(\mathbf{Y} + \mathbf{Z}) = \mathbf{\Sigma_Y} + \mathbf{\Sigma_{YZ}} + \mathbf{\Sigma_{ZY}} + \mathbf{\Sigma_Z}$$

### A.6 Tanh layers †

We use a linear combination of **independent** error functions with different scaling factors to approximate **tanh** function. In our experiments, we choose a set of four scaling parameters, $\{0.5583, 0.8596, 0.8596, 1.2612\}$, using `fmincon` in MATLAB. In practice, one can add more terms for even higher accuracy without losing efficiency (depending on the computing resources), because the extra terms can be easily paralleled. We define a variance term considering the relation between error function and Gaussian cdf, such that

---

† For notational simplicity, all the product, division, and power operations that appear in and after this section are all element-wise.

$$\mathbf{tanh}(\mathbf{X}) \approx \frac{1}{p} \sum_{j=1}^{p} \mathbf{erf}\left(\gamma_j \mathbf{X}\right)$$

$$\acute{\sigma}_j^2 = \frac{1}{2\gamma_j^2} \tag{6}$$

Thus, the factors of the pseudo Taylor polynomials are

$$\boldsymbol{A}_0 = \mathbb{E}\left[\frac{1}{p}\sum_{j=1}^{p}\mathbf{erf}\left(\frac{\mathbf{X}}{\sqrt{2\acute{\sigma}_j^2}}\right)\right]$$

$$\boldsymbol{A}_1 = \mathbb{E}\left[\nabla_{\mathbf{x}}\left(\frac{1}{p}\sum_{j=1}^{p}\mathbf{erf}\left(\frac{\mathbf{X}}{\sqrt{2\acute{\sigma}_j^2}}\right)\right)\right]$$

$$\boldsymbol{A}_2 = \frac{1}{2!}\mathbb{E}\left[\nabla_{\mathbf{x}}^2\left(\frac{1}{p}\sum_{j=1}^{p}\mathbf{erf}\left(\frac{\mathbf{X}}{\sqrt{2\acute{\sigma}_j^2}}\right)\right)\right]$$

$$\cdots$$

Since all the operations in $\boldsymbol{A}_0, \boldsymbol{A}_1, \boldsymbol{A}_2, \cdots$ are element-wise, we only show the derivation for univariate case for notational simplicity in the following sections

### A.6.1 Find $A_0$

$$\mathbb{E}\left[\mathbf{erf}\left(\frac{\mathrm{X}}{\sqrt{2\acute{\sigma}_j^2}}\right)\right] = \int_{-\infty}^{\infty}\mathbf{erf}\left(\frac{x}{\sqrt{2\acute{\sigma}_j^2}}\right)\frac{1}{\tilde{\sigma}}\varphi\left(\frac{x-\tilde{\mu}}{\tilde{\sigma}}\right)\mathrm{d}x$$

This is a known integral Ng and Geller (1969)

$$= \mathbf{erf}\left(\frac{\tilde{\mu}}{\sqrt{2\tilde{\sigma}^2 + 2\acute{\sigma}_j^2}}\right)$$

We define

$$\hat{\sigma}_j^2 = \tilde{\sigma}^2 + \acute{\sigma}_j^2 \tag{7}$$

Thus,

$$\boxed{A_0 = \frac{1}{p}\sum_{j=1}^{p}\mathbf{erf}\left(\frac{z}{\sqrt{2\hat{\sigma}_j^2}}\right)} \tag{8}$$

The usage of error function instead of Gaussian cdf may give $A_0$ a very distinctive form from those of the other factors. The reasons behind are purely out of considerations of numerical computing: calculating Gaussian cdf is computationally demanding, while the approximation algorithm of the error function is available Cody (1969).

### A.6.2 Find $A_1$

Notice that

$$\nabla\mathbf{erf}\left(\frac{\mathrm{X}}{\sqrt{2\acute{\sigma}_j^2}}\right) = \frac{\partial}{\partial x}\left(\int_0^{x/\sqrt{2\acute{\sigma}_j^2}}\frac{2}{\sqrt{\pi}}\mathbf{exp}(-t^2)\mathrm{d}t\right)$$

by Leibniz integral rule

$$= \frac{2}{\sqrt{\pi}} \frac{1}{\sqrt{2\acute{\sigma}_j^2}} \exp\left(-\frac{x^2}{2\acute{\sigma}_j^2}\right)$$

$$= \frac{2}{\acute{\sigma}_j} \varphi\left(\frac{x}{\acute{\sigma}_j}\right)$$

where $\varphi$ is the standard normal pdf. With the identity that the convolution of two Gaussians is still a Gaussian. (Bromiley (2003))

$$\mathbb{E}\left[\nabla \mathbf{erf}\left(\frac{X}{\sqrt{2\acute{\sigma}_j^2}}\right)\right] = \int_{-\infty}^{\infty} \frac{2}{\acute{\sigma}_j} \varphi\left(\frac{x}{\acute{\sigma}_j}\right) \frac{1}{\tilde{\sigma}} \varphi\left(\frac{x - \tilde{\mu}}{\tilde{\sigma}}\right) dx$$

$$= \frac{2}{\sqrt{\tilde{\sigma}^2 + \acute{\sigma}_j^2}} \varphi\left(\frac{\tilde{\mu}}{\sqrt{\tilde{\sigma}^2 + \acute{\sigma}_j^2}}\right)$$

Therefore,

$$\boxed{A_1 = \frac{1}{p}\sum_{j=1}^{p} \frac{2}{\hat{\sigma}_j} \varphi\left(\frac{\tilde{\mu}}{\hat{\sigma}_j}\right)} \tag{9}$$

and each term of the summation is a Gaussian function written in its standardized form.

### A.6.3   Find $A_2$ and beyond

In previous section, we show that the first derivative of the error function is a Gaussian, thus the expected value of which is the convolution of two Gaussians. Similarly, we can obtain $A_2$, $A_3$, etc. by convolving the second, third, and higher order Gaussian derivatives with another Gaussian.

Gaussian derivatives can be represented by Hermite polynomial $\mathbf{H}_s(x)$.

$$\mathbf{H}_0(x) = 1$$
$$\mathbf{H}_1(x) = 2x$$
$$\mathbf{H}_2(x) = 4x^2 - 2$$
$$\mathbf{H}_3(x) = 8x^3 - 12x$$
$$\cdots$$

There are implemented functions for this from various scientific computing tools, such as hermiteH() from MATLAB and scipy.special.hermite() from SciPy.

$$\frac{d^s}{dx^s}\left[\frac{1}{\sigma}\varphi\left(\frac{x}{\sigma}\right)\right] = \left(\frac{-1}{\sqrt{2\sigma^2}}\right)^s \mathbf{H}_s\left(\frac{x}{\sqrt{2\sigma^2}}\right) \frac{1}{\sigma}\varphi\left(\frac{x}{\sigma}\right)$$

Hence,

$$\mathbb{E}\left[\nabla^s \mathbf{erf}\left(\frac{\mathrm{X}}{\sqrt{2\acute{\sigma}_j^2}}\right)\right]$$

$$=\int_{-\infty}^{\infty}\left[\frac{\partial^s}{\partial x^s}\mathbf{erf}\left(\frac{x}{\sqrt{2\acute{\sigma}_j^2}}\right)\right]p(x)\mathrm{d}x$$

$$=2\int_{-\infty}^{\infty}\frac{\partial^{s-1}}{\partial x^{s-1}}\left[\frac{1}{\acute{\sigma}_j}\varphi\left(\frac{x}{\acute{\sigma}_j}\right)\right]\frac{1}{\tilde{\sigma}}\varphi\left(\frac{x-\tilde{\mu}}{\tilde{\sigma}}\right)\mathrm{d}x$$

$$=2\left(\frac{-1}{\sqrt{2\acute{\sigma}_j^2}}\right)^{s-1}\int_{-\infty}^{\infty}\mathbf{H}_{s-1}\left(\frac{x}{\sqrt{2\acute{\sigma}_j^2}}\right)\frac{1}{\acute{\sigma}_j}\varphi\left(\frac{x}{\acute{\sigma}_j}\right)\frac{1}{\tilde{\sigma}}\varphi\left(\frac{x-\tilde{\mu}}{\tilde{\sigma}}\right)dx$$

$$=2\left(\frac{-1}{\sqrt{2\acute{\sigma}_j^2}}\right)^{s-1}\frac{1}{\hat{\sigma}_j}\varphi\left(\frac{\bar{\mu}}{\hat{\sigma}_j}\right)\int_{-\infty}^{\infty}\mathbf{H}_{s-1}\left(\frac{x}{\sqrt{2\acute{\sigma}_j^2}}\right)\frac{1}{\bar{\sigma}_j}\varphi\left(\frac{x-\bar{\mu}}{\bar{\sigma}_j}\right)dx$$

where $\bar{\mu}=\tilde{\mu}\frac{\acute{\sigma}_j^2}{\hat{\sigma}_j^2}\quad\bar{\sigma}_j^2=\tilde{\sigma}^2\frac{\acute{\sigma}_j^2}{\hat{\sigma}_j^2}$. The convolution of a Hermite polynomial and a Gaussian pdf is a known integral Gradshteyn and Ryzhik (2015)

$$=2\left(\frac{-1}{\sqrt{2\acute{\sigma}_j^2}}\right)^{s-1}\frac{1}{\hat{\sigma}_j}\varphi\left(\frac{\tilde{\mu}}{\hat{\sigma}_j}\right)\left(1-2\bar{\sigma}^2\frac{1}{2\acute{\sigma}_j^2}\right)^{\frac{s-1}{2}}\mathbf{H}_{s-1}\left(\frac{\bar{\mu}/\sqrt{2\acute{\sigma}_j^2}}{\left(1-2\bar{\sigma}^2\frac{1}{2\acute{\sigma}_j^2}\right)^{\frac{1}{2}}}\right)$$

$$=2\left(\frac{-1}{\sqrt{2\acute{\sigma}_j^2}}\right)^{s-1}\mathbf{H}_{s-1}\left(\frac{\tilde{\mu}}{\sqrt{2\hat{\sigma}_j^2}}\right)\frac{1}{\hat{\sigma}_j}\varphi\left(\frac{\tilde{\mu}}{\hat{\sigma}_j}\right)$$

Therefore, we can write the formula of $A_s$ for $s\geq 1$

$$\boxed{A_s(s\geq 1)=\frac{1}{s!}\frac{1}{p}\sum_{j=1}^{p}2\left(\frac{-1}{\sqrt{2\hat{\sigma}_j^2}}\right)^{s-1}\mathbf{H}_{s-1}\left(\frac{\tilde{\mu}}{\sqrt{2\hat{\sigma}_j^2}}\right)\frac{1}{\hat{\sigma}_j}\varphi\left(\frac{\tilde{\mu}}{\hat{\sigma}_j}\right)} \tag{10}$$

To give a few examples,

$$\boxed{\begin{aligned}A_2&=\frac{1}{2!}\frac{1}{p}\sum_{j=1}^{p}-2\frac{\tilde{\mu}}{\hat{\sigma}_j^2}\frac{1}{\hat{\sigma}_j}\varphi\left(\frac{\tilde{\mu}}{\hat{\sigma}_j}\right)\\A_3&=\frac{1}{3!}\frac{1}{p}\sum_{j=1}^{p}2\frac{\tilde{\mu}^2-\hat{\sigma}_j^2}{\hat{\sigma}_j^4}\frac{1}{\hat{\sigma}_j}\varphi\left(\frac{\tilde{\mu}}{\hat{\sigma}_j}\right)\\A_4&=\frac{1}{4!}\frac{1}{p}\sum_{j=1}^{p}2\left(\frac{-\tilde{\mu}^3+3\tilde{\mu}\hat{\sigma}_j^2}{\hat{\sigma}_j^6}\right)\frac{1}{\hat{\sigma}_j}\varphi\left(\frac{\tilde{\mu}}{\hat{\sigma}_j}\right)\\\dots\end{aligned}} \tag{11}$$

Note that we will reuse this relation in the following section

$$\int_{-\infty}^{\infty}\left(\frac{-1}{\sqrt{2\acute{\sigma}_j^2}}\right)^s\mathbf{H}_s\left(\frac{x}{\sqrt{2\acute{\sigma}_j^2}}\right)\frac{1}{\acute{\sigma}_j}\varphi\left(\frac{x}{\acute{\sigma}_j}\right)\frac{1}{\tilde{\sigma}}\varphi\left(\frac{x-\tilde{\mu}}{\tilde{\sigma}}\right)dx=\left(\frac{-1}{\sqrt{2\hat{\sigma}_j^2}}\right)^s\mathbf{H}_s\left(\frac{\tilde{\mu}}{\sqrt{2\hat{\sigma}_j^2}}\right)\frac{1}{\hat{\sigma}_j}\varphi\left(\frac{\tilde{\mu}}{\hat{\sigma}_j}\right) \tag{12}$$

## A.7 Sigmoid layers

We can apply the same framework on **sigmoid** layers, with modifications

$$\mathbf{sigmoid}(u+b) = \frac{1}{1+\mathbf{exp}(-(u+b))} \approx \frac{1}{2} + \frac{1}{2p}\sum_{j=1}^{p}\mathbf{erf}\left[\gamma_j(u+b))\right]$$

Using `fmincon` in MATLAB, we find a set of $\gamma = (0.2791, 0.4298, 0.4298, 0.6306)^\top$. Then the first four factors of the Taylor polynomials are listed below. $A_0$ is represented in complementary error function **erfc** to avoid subtractive cancellation that leads to inaccuracy in the tails. Note that except for $A_0$, all $A_s$ of **sigmoid** layers are just $1/2$ of those of **tanh** layers.

$$
\begin{aligned}
A_0 &= \frac{1}{p}\sum_{j=1}^{p}\frac{1}{2}\mathbf{erfc}\left(-\frac{\tilde{\mu}}{\sqrt{2\hat{\sigma}_j^2}}\right) \\
A_1 &= \frac{1}{p}\sum_{j=1}^{p}\frac{1}{\hat{\sigma}_j}\,\varphi\left(\frac{\tilde{\mu}}{\hat{\sigma}_j}\right) \\
A_2 &= \frac{1}{2!\,p}\sum_{j=1}^{p}-\frac{\tilde{\mu}}{\hat{\sigma}_j^2}\frac{1}{\hat{\sigma}_j}\varphi\left(\frac{\tilde{\mu}}{\hat{\sigma}_j}\right) \\
A_3 &= \frac{1}{3!\,p}\sum_{j=1}^{p}\frac{\tilde{\mu}^2-\hat{\sigma}_j^2}{\hat{\sigma}_j^4}\frac{1}{\hat{\sigma}_j}\varphi\left(\frac{\tilde{\mu}}{\hat{\sigma}_j}\right) \\
A_4 &= \frac{1}{4!\,p}\sum_{j=1}^{p}\left(\frac{-\tilde{\mu}^3+3\tilde{\mu}\hat{\sigma}_j^2}{\hat{\sigma}_j^6}\right)\frac{1}{\hat{\sigma}_j}\varphi\left(\frac{\tilde{\mu}}{\hat{\sigma}_j}\right) \\
&\dots
\end{aligned}
\tag{13}
$$

## A.8 Softplus layers

The derivation of pseudo-Taylor polynomials for a **softplus** layer is related to that for a **sigmoid** layer, since the derivative of the **softplus** function is the **sigmoid** function with scaling factor $\beta$, and the latter can be approximated with a linear combination of Gaussian cdf (or error functions like we did in the previous section). We have

$$\mathbf{softplus}(x) = \frac{1}{\beta}\mathbf{log}\left(1+e^{\beta x}\right)$$

Then we use the approximation of sum of **independent** standard Gaussian cdf $\mathbf{\Phi}$

$$\frac{\partial}{\partial x}\mathbf{softplus}(x) = \frac{1}{1+e^{-\beta x}} \approx \frac{1}{p}\sum_{j=1}^{p}\mathbf{\Phi}\left(\frac{x}{\acute{\sigma}_j}\right) \tag{14}$$

where we re-define

$$\acute{\sigma}_j^2 = \frac{1}{2\gamma_j^2\beta^2} \tag{15}$$

Note that $\acute{\sigma}_j^2$ changes definition and should not be confused with that in the **tanh** and **sigmoid** sections.

### A.8.1 Find $A_0$

First we apply substitution of variables $X = \tilde{\mu} + \Xi$, then

$$\mathbf{softplus}(x) = \mathbf{softplus}(\tilde{\mu}, \xi) = \frac{1}{\beta}\mathbf{log}\left(1+e^{\beta(\tilde{\mu}+\xi)}\right)$$

Notice that $\dfrac{\partial}{\partial x} = \dfrac{\partial}{\partial \xi}$ since $\tilde{\mu}$ is constant, then

$$
\begin{aligned}
A_0 &= \int_{-\infty}^{\infty} \mathbf{softplus}(\tilde{\mu}, \xi)\; p(\xi)\; \mathrm{d}\xi \\
&= \int_{-\infty}^{\infty} p(\xi)\mathrm{d}\xi \int_{-\infty}^{\tilde{\mu}} \frac{\partial}{\partial \zeta}\mathbf{softplus}(\zeta, \xi)\; \mathrm{d}\zeta
\end{aligned}
$$

by Fubini's theorem

$$
\begin{aligned}
&= \int_{-\infty}^{\tilde{\mu}} \mathrm{d}\zeta \int_{-\infty}^{\infty} \frac{\partial}{\partial \zeta}\mathbf{softplus}(\zeta, \xi)\; p(\xi)\mathrm{d}\xi \\
&\approx \frac{1}{p}\sum_{j=1}^{p} \int_{-\infty}^{\tilde{\mu}} \mathrm{d}\zeta \int_{-\infty}^{\infty} \mathbf{\Phi}\left(\frac{\zeta + \xi}{\acute{\sigma}_j}\right) \frac{1}{\tilde{\sigma}}\boldsymbol{\varphi}\left(\frac{\xi}{\tilde{\sigma}}\right) \mathrm{d}\xi
\end{aligned}
$$

the convolution of Gaussian cdf and pdf, the integral of Gaussian cdf are known integrals

$$
\begin{aligned}
&= \frac{1}{p}\sum_{j=1}^{p} \int_{-\infty}^{\tilde{\mu}} \mathbf{\Phi}\left(\frac{\zeta}{\hat{\sigma}_j}\right) \mathrm{d}\zeta \\
&= \frac{1}{p}\sum_{j=1}^{p} \left[\tilde{\mu}\; \mathbf{\Phi}\left(\frac{\tilde{\mu}}{\hat{\sigma}_j}\right) + \hat{\sigma}_j\boldsymbol{\varphi}\left(\frac{\tilde{\mu}}{\hat{\sigma}_j}\right)\right] \\
&= \frac{1}{p}\sum_{j=1}^{p} \left[\frac{\tilde{\mu}}{2}\; \mathbf{erfc}\left(-\frac{\tilde{\mu}}{\sqrt{2\hat{\sigma}_j^2}}\right) + \hat{\sigma}_j\boldsymbol{\varphi}\left(\frac{\tilde{\mu}}{\hat{\sigma}_j}\right)\right]
\end{aligned}
$$

Or, with simplification

$$
\boxed{A_0 = A_1\tilde{\mu} + \frac{1}{p}\sum_{j=1}^{p} \hat{\sigma}_j\boldsymbol{\varphi}\left(\frac{\tilde{\mu}}{\hat{\sigma}_j}\right)} \tag{16}
$$

### A.8.2   Find $A_1$

Since the first derivative of the **softplus** function is just a **sigmoid** function with scaling factor $\beta$, we can immediately write $A_1$ using previous results

$$
\boxed{A_1 = \frac{1}{p}\sum_{j=1}^{p} \frac{1}{2}\mathbf{erfc}\left(-\frac{\tilde{\mu}}{\sqrt{2\hat{\sigma}_j^2}}\right)} \tag{17}
$$

### A.8.3   Find $A_2$ and beyond

In previous section, we find that $\nabla\,\mathbf{softplus}(x)$ is approximately a Gaussian cdf. Subsequently, $\nabla^2\,\mathbf{softplus}(x)$ is approximately a Gaussian. Since Gaussian function is infinitely differentiable, all $A_s(s > 2)$ can be found

using Gaussian derivatives, which can be represented by Hermite polynomial $\mathbf{H}_s(x)$ introduced above.

$$
\begin{aligned}
A_s &= \frac{1}{s!}\mathbb{E}\left[\frac{\partial^s}{\partial x^s}\mathbf{softplus}(x)\right]\\
&\approx \frac{1}{s!\,p}\sum_{j=1}^{p}\int_{-\infty}^{\infty}\frac{\partial^{s-2}}{\partial x^{s-2}}\left[\frac{1}{\acute{\sigma}_j}\varphi\left(\frac{x}{\acute{\sigma}_j}\right)\right]p(x)\,\mathrm{d}x\\
&= \frac{1}{s!\,p}\sum_{j=1}^{p}\left(\frac{-1}{\sqrt{2\acute{\sigma}_j^2}}\right)^{s-2}\int_{-\infty}^{\infty}\mathbf{H}_{s-2}\left(\frac{x}{\sqrt{2\acute{\sigma}_j^2}}\right)\frac{1}{\acute{\sigma}_j}\varphi\left(\frac{x}{\acute{\sigma}_j}\right)\frac{1}{\tilde{\sigma}}\varphi\left(\frac{x-\tilde{\mu}}{\tilde{\sigma}}\right)\mathrm{d}x
\end{aligned}
$$

we solved this integral in tanh section

$$
= \frac{1}{s!\,p}\sum_{j=1}^{p}\left(\frac{-1}{\sqrt{2\hat{\sigma}_j^2}}\right)^{s-2}\mathbf{H}_{s-2}\left(\frac{\tilde{\mu}}{\sqrt{2\hat{\sigma}_j^2}}\right)\frac{1}{\hat{\sigma}_j}\varphi\left(\frac{\tilde{\mu}}{\hat{\sigma}_j}\right)
$$

To summarize, $A_s(s \geq 2)$ can be expressed as

$$
A_s(s \geq 2) = \frac{1}{s!\,p}\sum_{j=1}^{p}\left(\frac{-1}{\sqrt{2\hat{\sigma}_j^2}}\right)^{s-2}\mathbf{H}_{s-2}\left(\frac{\tilde{\mu}}{\sqrt{2\hat{\sigma}_j^2}}\right)\frac{1}{\hat{\sigma}_j}\varphi\left(\frac{\tilde{\mu}}{\hat{\sigma}_j}\right) \tag{18}
$$

For examples,

$$
\begin{aligned}
A_2 &= \frac{1}{2!\,p}\sum_{j=1}^{p}\frac{1}{\hat{\sigma}_j}\varphi\left(\frac{\tilde{\mu}}{\hat{\sigma}_j}\right)\\
A_3 &= \frac{1}{3!\,p}\sum_{j=1}^{p}-\frac{\tilde{\mu}}{\hat{\sigma}_j^2}\frac{1}{\hat{\sigma}_j}\varphi\left(\frac{\tilde{\mu}}{\hat{\sigma}_j}\right)\\
A_4 &= \frac{1}{4!\,p}\sum_{j=1}^{p}\frac{\tilde{\mu}^2-\hat{\sigma}_j^2}{\hat{\sigma}_j^4}\frac{1}{\hat{\sigma}_j}\varphi\left(\frac{\tilde{\mu}}{\hat{\sigma}_j}\right)\\
&\dots
\end{aligned} \tag{19}
$$

### A.9  ReLU, Leaky ReLU, and Piece-wise Linear layers

Since ReLU function is only first-order differentiable ($x > 0$), we cannot do PTPE directly. However, given its relation to **softplus** function,

$$
\lim_{\beta\to\infty}\frac{1}{\beta}\mathbf{log}\left(1+e^{\beta x}\right) = \max\{0,x\}
$$

we can reuse the results for **softplus** layers by applying the limit

$$
\lim_{\beta\to\infty}\acute{\sigma}_j^2 = 0 \qquad \text{and} \qquad \lim_{\beta\to\infty}\hat{\sigma}_j^2 = \tilde{\sigma}^2
$$

Therefore,

$$
\begin{aligned}
A_0 &= A_1\tilde{\mu} + \tilde{\sigma}\boldsymbol{\varphi}\left(\frac{\tilde{\mu}}{\tilde{\sigma}}\right) \\
A_1 &= \frac{1}{2}\mathbf{erfc}\left(-\frac{\tilde{\mu}}{\sqrt{2\tilde{\sigma}^2}}\right) \\
A_2 &= \frac{1}{2!}\frac{1}{\tilde{\sigma}}\ \boldsymbol{\varphi}\left(\frac{\tilde{\mu}}{\tilde{\sigma}}\right) \\
A_3 &= \frac{1}{3!} - \frac{\tilde{\mu}}{\tilde{\sigma}^2}\frac{1}{\tilde{\sigma}}\boldsymbol{\varphi}\left(\frac{\tilde{\mu}}{\tilde{\sigma}}\right) \\
A_4 &= \frac{1}{4!}\frac{\tilde{\mu}^2 - \tilde{\sigma}^2}{\tilde{\sigma}^4}\frac{1}{\tilde{\sigma}}\boldsymbol{\varphi}\left(\frac{\tilde{\mu}}{\tilde{\sigma}}\right) \\
&\dots
\end{aligned}
\tag{20}
$$

and for $s \geq 2$ we have the general form of

$$
A_s(s \geq 2) = \frac{1}{s!}\left(\frac{-1}{\sqrt{2\tilde{\sigma}_j^2}}\right)^{s-2}\mathbf{H}_{s-2}\left(\frac{\tilde{\mu}}{\sqrt{2\tilde{\sigma}_j^2}}\right)\frac{1}{\tilde{\sigma}_j}\boldsymbol{\varphi}\left(\frac{\tilde{\mu}}{\tilde{\sigma}_j}\right)
\tag{21}
$$

On the other hand, leaky ReLU can be considered as superposition of two ReLU functions - consider a leaky ReLU with negative slope of $\theta$

$$
\mathbf{LeakyReLU}(x;\theta) = \mathbf{ReLU}(x) - \theta\ \mathbf{ReLU}(-x)
\tag{22}
$$

which can also be written as

$$
\lim_{\beta\to\infty}\mathbf{softplus}(x) - \theta\ \mathbf{softplus}(-x)
$$

Therefore,

$$
\begin{aligned}
A_0 &= \lim_{\beta\to\infty}\frac{1}{p}\sum_{j=1}^{p}\left[\tilde{\mu}\boldsymbol{\Phi}\left(\frac{\tilde{\mu}}{\hat{\sigma}_j}\right) + \hat{\sigma}_j\boldsymbol{\varphi}\left(\frac{\tilde{\mu}}{\hat{\sigma}_j}\right)\right] - \theta\left[-\tilde{\mu}\boldsymbol{\Phi}\left(-\frac{\tilde{\mu}}{\hat{\sigma}_j}\right) + \hat{\sigma}_j\boldsymbol{\varphi}\left(\frac{\tilde{\mu}}{\hat{\sigma}_j}\right)\right] \\
&= \theta\tilde{\mu} + (1-\theta)\left[\tilde{\mu}\boldsymbol{\Phi}\left(\frac{\tilde{\mu}}{\tilde{\sigma}}\right) + \tilde{\sigma}\boldsymbol{\varphi}\left(\frac{\tilde{\mu}}{\tilde{\sigma}}\right)\right]
\end{aligned}
$$

To find the expected value of the derivative of **LeakyReLU**, first we find the derivative

$$
\begin{aligned}
\frac{\partial}{\partial x}\mathbf{LeakyReLU}(x\ ;\ \theta) &= \lim_{\beta\to\infty}\frac{\partial}{\partial x}\mathbf{softplus}(x) - \theta\frac{\partial}{\partial x}\mathbf{softplus}(-(x)) \\
&= \lim_{\beta\to\infty}\frac{1}{1+e^{-\beta(x)}} + \frac{\theta}{1+e^{\beta(x)}} \\
&\approx \frac{1}{p}\sum_{j=1}^{p}\boldsymbol{\Phi}\left(\frac{x}{\acute{\sigma}_j}\right) + \theta\boldsymbol{\Phi}\left(\frac{-x}{\acute{\sigma}_j}\right) \\
&= \lim_{\beta\to\infty}\theta + \frac{1-\theta}{p}\sum_{j=1}^{p}\boldsymbol{\Phi}\left(\frac{x}{\acute{\sigma}_j}\right)
\end{aligned}
$$

Then we can write $A_1$ for **LeakyReLU** as

$$A_1 = \lim_{\beta \to \infty} \int_{-\infty}^{\infty} \left[ \theta + \frac{1-\theta}{p} \sum_{j=1}^{p} \mathbf{\Phi}\left(\frac{x}{\acute{\sigma}_j}\right) \right] \frac{1}{\tilde{\sigma}} \varphi\left(\frac{x - \tilde{\mu}}{\tilde{\sigma}}\right) \mathrm{d}x$$

$$= \theta + \lim_{\beta \to \infty} \frac{1-\theta}{p} \sum_{j=1}^{p} \int_{-\infty}^{\infty} \mathbf{\Phi}\left(\frac{x}{\acute{\sigma}_j}\right) \frac{1}{\tilde{\sigma}} \varphi\left(\frac{x - \tilde{\mu}}{\tilde{\sigma}}\right) \mathrm{d}x$$

$$= \theta + \lim_{\beta \to \infty} \frac{1-\theta}{p} \sum_{j=1}^{p} \mathbf{\Phi}\left(\frac{\tilde{\mu}}{\hat{\sigma}_j}\right)$$

$$= \theta + (1-\theta)\mathbf{\Phi}\left(\frac{\tilde{\mu}}{\tilde{\sigma}}\right)$$

Rewrite in complementary error function

$$A_1 = \theta + \frac{1-\theta}{2} \mathbf{erfc}\left(-\frac{\tilde{\mu}}{\sqrt{2\tilde{\sigma}^2}}\right) \tag{23}$$

Note that we can also rewrite $A_0$ using the result of $A_1$ to improve computational efficiency.

$$A_0 = A_1 \tilde{\mu} + (1-\theta)\tilde{\sigma} \varphi\left(\frac{\tilde{\mu}}{\tilde{\sigma}}\right) \tag{24}$$

Note that starting from the second order, the derivative of **LeakyReLU** is just that of **ReLU** scaled by $1 - \theta$. Therefore,

$$\begin{aligned} A_2 &= \frac{1-\theta}{2} \frac{1}{\tilde{\sigma}} \varphi\left(\frac{\tilde{\mu}}{\tilde{\sigma}}\right) \\ A_3 &= -\frac{1-\theta}{3!} \frac{\tilde{\mu}}{\tilde{\sigma}^3} \varphi\left(\frac{\tilde{\mu}}{\tilde{\sigma}}\right) \\ A_4 &= \frac{1-\theta}{4!} \frac{\tilde{\mu}^2 - \tilde{\sigma}^2}{\tilde{\sigma}^5} \varphi\left(\frac{\tilde{\mu}}{\tilde{\sigma}}\right) \\ &\dots \end{aligned} \tag{25}$$

and for $s \geq 2$, we have the general form of

$$A_s(s \geq 2) = \frac{1-\theta}{s!} \left(\frac{-1}{\sqrt{2\tilde{\sigma}^2}}\right)^{s-2} \mathbf{H}_{s-2}\left(\frac{\tilde{\mu}}{\sqrt{2\tilde{\sigma}^2}}\right) \frac{1}{\tilde{\sigma}} \varphi\left(\frac{\tilde{\mu}}{\tilde{\sigma}}\right) \tag{26}$$

Similarly, any piece-wise linear activation function can be described as a combination of ReLU functions with different scaling, shifting, and/or mirroring. Thus, their pseudo Taylor coefficients can be found using the same methodology.

### A.10 GELU layers

**GELU** (Gaussian Error Linear Unit) is defined as the product of input and a standard Gaussian cdf

$$\mathbf{GELU}(x) = x \ \mathbf{\Phi}(x)$$

and we can write the derivatives (with order $s \geq 1$) of **GELU** as

$$\frac{\partial^s}{\partial x^s} \mathbf{GELU}(x) = s \frac{\partial^{s-1}}{\partial x^{s-1}} \mathbf{\Phi}(x) + x \frac{\partial^s}{\partial x^s} \mathbf{\Phi}(x)$$

### A.10.1 Find $A_0$

$$A_0 = \mathbb{E}\left[\mathbf{GELU}(x)\right]$$

$$= \int_{-\infty}^{\infty} x\mathbf{\Phi}(x)\frac{1}{\tilde{\sigma}}\boldsymbol{\varphi}\left(\frac{x-\tilde{\mu}}{\tilde{\sigma}}\right)\mathrm{d}x$$

$$= \int_{-\infty}^{\infty}(\tilde{\mu}+\xi)\int_{-\infty}^{\tilde{\mu}}\boldsymbol{\varphi}\left(\zeta+\xi\right)\mathrm{d}\zeta\,\frac{1}{\tilde{\sigma}}\boldsymbol{\varphi}\left(\frac{\xi}{\tilde{\sigma}}\right)\mathrm{d}\xi$$

$$= \tilde{\mu}\int_{-\infty}^{\tilde{\mu}}\int_{-\infty}^{\infty}\boldsymbol{\varphi}\left(\zeta+\xi\right)\frac{1}{\tilde{\sigma}}\boldsymbol{\varphi}\left(\frac{\xi}{\tilde{\sigma}}\right)\mathrm{d}\xi\,\mathrm{d}\zeta\;+\int_{-\infty}^{\tilde{\mu}}\int_{-\infty}^{\infty}\xi\,\boldsymbol{\varphi}\left(\zeta+\xi\right)\frac{1}{\tilde{\sigma}}\boldsymbol{\varphi}\left(\frac{\xi}{\tilde{\sigma}}\right)\mathrm{d}\xi\,\mathrm{d}\zeta$$

$$= \tilde{\mu}\int_{-\infty}^{\tilde{\mu}}\frac{1}{\sqrt{1+\tilde{\sigma}^2}}\boldsymbol{\varphi}\left(\frac{\zeta}{\sqrt{1+\tilde{\sigma}^2}}\right)\mathrm{d}\zeta\;+\int_{-\infty}^{\tilde{\mu}}\frac{1}{\sqrt{1+\tilde{\sigma}^2}}\boldsymbol{\varphi}\left(\frac{\zeta}{\sqrt{1+\tilde{\sigma}^2}}\right)\frac{-\zeta\tilde{\sigma}^2}{1+\tilde{\sigma}^2}\mathrm{d}\zeta$$

$$= \tilde{\mu}\mathbf{\Phi}\left(\frac{\tilde{\mu}}{\sqrt{1+\tilde{\sigma}^2}}\right)+\frac{\tilde{\sigma}^2}{\sqrt{1+\tilde{\sigma}^2}}\boldsymbol{\varphi}\left(\frac{\tilde{\mu}}{\sqrt{1+\tilde{\sigma}^2}}\right)$$

We re-define $\hat{\sigma}^2$

$$\hat{\sigma}^2 = 1+\tilde{\sigma}^2 \tag{27}$$

and re-write the result with complementary error function

$$\boxed{A_0 = \frac{\tilde{\mu}}{2}\mathbf{erfc}\left(-\frac{\tilde{\mu}}{\sqrt{2\hat{\sigma}^2}}\right)+\frac{\tilde{\sigma}^2}{\hat{\sigma}}\boldsymbol{\varphi}\left(\frac{\tilde{\mu}}{\hat{\sigma}}\right)} \tag{28}$$

### A.10.2 Find $A_1$

$$A_1 = \mathbb{E}\left[\frac{\partial}{\partial x}\mathbf{GELU}(x)\right]$$

$$= \int_{-\infty}^{\infty}\mathbf{\Phi}(x)\frac{1}{\tilde{\sigma}}\boldsymbol{\varphi}(\frac{x-\tilde{\mu}}{\tilde{\sigma}})\mathrm{d}x\;+\int_{-\infty}^{\infty}x\boldsymbol{\varphi}(x)\frac{1}{\tilde{\sigma}}\boldsymbol{\varphi}(\frac{x-\tilde{\mu}}{\tilde{\sigma}})\mathrm{d}x$$

using results of previous section

$$= \mathbf{\Phi}\left(\frac{\tilde{\mu}}{\hat{\sigma}}\right)+\frac{\tilde{\mu}}{\hat{\sigma}^2}\frac{1}{\hat{\sigma}}\boldsymbol{\varphi}\left(\frac{\tilde{\mu}}{\hat{\sigma}}\right)$$

Therefore,

$$\boxed{A_1 = \frac{1}{2}\mathbf{erfc}\left(-\frac{\tilde{\mu}}{\sqrt{2\hat{\sigma}^2}}\right)+\frac{\tilde{\mu}}{\hat{\sigma}^2}\frac{1}{\hat{\sigma}}\boldsymbol{\varphi}\left(\frac{\tilde{\mu}}{\hat{\sigma}}\right)} \tag{29}$$

### A.10.3 Find $A_2$ and beyond

Higher order coefficients $(A_s(s \geq 2))$ all consist of two parts: (i) a term of expected value of a Gaussian derivative, (ii) a term of expected value of the product of $x$ and a Gaussian derivative. We have already found a general form of the first term in the **tanh** section

$$\mathbb{E}\left[s\frac{\partial^{s-2}}{\partial x^{s-2}}\boldsymbol{\varphi}(x)\right] = s\left(\frac{-1}{\sqrt{2\hat{\sigma}^2}}\right)^{s-2}\mathbf{H}_{s-2}\left(\frac{\tilde{\mu}}{\sqrt{2\hat{\sigma}^2}}\right)\frac{1}{\hat{\sigma}}\boldsymbol{\varphi}\left(\frac{\tilde{\mu}}{\hat{\sigma}}\right)$$

To solve the second part, we need to use the Hermite polynomial recurrence relation:

$$x\,\mathbf{H}_{s-1}(x) = \frac{1}{2}\mathbf{H}_s(x)+s\,\mathbf{H}_{s-2}(x)$$

$$\mathbb{E}\left[x\frac{\partial^{s-1}}{\partial x^{s-1}}\boldsymbol{\varphi}(x)\right]$$

$$= \left(\frac{-1}{\sqrt{2}}\right)^{s-1}\int_{-\infty}^{\infty} x\;\mathbf{H}_{s-1}\left(\frac{x}{\sqrt{2}}\right)\boldsymbol{\varphi}\left(x\right)\frac{1}{\tilde{\sigma}}\boldsymbol{\varphi}\left(\frac{x-\tilde{\mu}}{\tilde{\sigma}}\right)\mathrm{d}x$$

$$= \left(\frac{-1}{\sqrt{2}}\right)^{s-1}\sqrt{2}\int_{-\infty}^{\infty} \frac{x}{\sqrt{2}}\;\mathbf{H}_{s-1}\left(\frac{x}{\sqrt{2}}\right)\;\boldsymbol{\varphi}\left(x\right)\frac{1}{\tilde{\sigma}}\boldsymbol{\varphi}\left(\frac{x-\tilde{\mu}}{\tilde{\sigma}}\right)\mathrm{d}x$$

$$= \left(\frac{-1}{\sqrt{2}}\right)^{s-1}\sqrt{2}\int_{-\infty}^{\infty}\left[\frac{1}{2}\mathbf{H}_s\left(\frac{x}{\sqrt{2}}\right)+(s-1)\mathbf{H}_{s-2}\left(\frac{x}{\sqrt{2}}\right)\right]\boldsymbol{\varphi}\left(x\right)\frac{1}{\tilde{\sigma}}\boldsymbol{\varphi}\left(\frac{x-\tilde{\mu}}{\tilde{\sigma}}\right)\mathrm{d}x$$

$$= -\left(\frac{-1}{\sqrt{2}}\right)^{s}\int_{-\infty}^{\infty}\mathbf{H}_s\left(\frac{x}{\sqrt{2}}\right)\boldsymbol{\varphi}\left(x\right)\frac{1}{\tilde{\sigma}}\boldsymbol{\varphi}\left(\frac{x-\tilde{\mu}}{\tilde{\sigma}}\right)\mathrm{d}x\;\cdots$$

$$-(s-1)\left(\frac{-1}{\sqrt{2}}\right)^{s-2}\int_{-\infty}^{\infty}\mathbf{H}_{s-2}\left(\frac{x}{\sqrt{2}}\right)\boldsymbol{\varphi}\left(x\right)\frac{1}{\tilde{\sigma}}\boldsymbol{\varphi}\left(\frac{x-\tilde{\mu}}{\tilde{\sigma}}\right)\mathrm{d}x$$

by equation 12

$$= \frac{1}{\hat{\sigma}}\boldsymbol{\varphi}\left(\frac{\tilde{\mu}}{\hat{\sigma}}\right)\left[-\left(\frac{-1}{\sqrt{2\hat{\sigma}^2}}\right)^{s}\mathbf{H}_s\left(\frac{\tilde{\mu}}{\sqrt{2\hat{\sigma}^2}}\right)-(s-1)\left(\frac{-1}{\sqrt{2\hat{\sigma}^2}}\right)^{s-2}\mathbf{H}_{s-2}\left(\frac{\tilde{\mu}}{\sqrt{2\hat{\sigma}^2}}\right)\right]$$

Sum the two integral together, we get the general form of $A_s(s \geq 2)$

$$A_s(s \geq 2) = \frac{1}{s!}\left[\left(\frac{-1}{\sqrt{2\hat{\sigma}^2}}\right)^{s-2}\mathbf{H}_{s-2}\left(\frac{\tilde{\mu}}{\sqrt{2\hat{\sigma}^2}}\right)-\left(\frac{-1}{\sqrt{2\hat{\sigma}^2}}\right)^{s}\mathbf{H}_s\left(\frac{\tilde{\mu}}{\sqrt{2\hat{\sigma}^2}}\right)\right]\frac{1}{\hat{\sigma}}\boldsymbol{\varphi}\left(\frac{\tilde{\mu}}{\hat{\sigma}}\right) \tag{30}$$

For examples,

$$A_2 = \frac{1}{2!}\left[1+\frac{1}{\hat{\sigma}^2}-\frac{\tilde{\mu}^2}{\hat{\sigma}^4}\right]\frac{1}{\hat{\sigma}}\boldsymbol{\varphi}\left(\frac{\tilde{\mu}}{\hat{\sigma}}\right)$$

$$A_3 = -\frac{1}{3!}\left[\frac{\tilde{\mu}}{\hat{\sigma}^2}+\frac{3\tilde{\mu}}{\hat{\sigma}^4}-\frac{\tilde{\mu}^3}{\hat{\sigma}^6}\right]\frac{1}{\hat{\sigma}}\boldsymbol{\varphi}\left(\frac{\tilde{\mu}}{\hat{\sigma}}\right) \tag{31}$$

$$A_4 = \frac{1}{4!}\left[-\frac{1}{\hat{\sigma}^2}+\frac{\tilde{\mu}^2-3}{\hat{\sigma}^4}+\frac{6\tilde{\mu}^2}{\hat{\sigma}^6}-\frac{\tilde{\mu}^4}{\hat{\sigma}^8}\right]\frac{1}{\hat{\sigma}}\boldsymbol{\varphi}\left(\frac{\tilde{\mu}}{\hat{\sigma}}\right)$$

$$\cdots$$

### A.11  SiLU layers

**SiLU** (Sigmoid Linear Unit), equivalent to **Swish** when $\beta = 1$, is defined as the product of input and a **sigmoid** function

$$\mathbf{SiLU}(x) = x\,\mathbf{Sigmoid}(x)$$

In the previous section, we approximate **Sigmoid** function with error functions so that we can reuse derivations from the **Tanh** section. Here we approximate **Sigmoid** function with Gaussian cdf's in order to reuse derivations from the **GELU** section. With $\gamma$ as a numerically optimized scalar vector, let

$$\acute{\sigma}_j^2 = \frac{1}{2\gamma_j^2}\quad,\quad j \in \{1,\cdots,p\}$$

Then, we approximate **SiLU** as

$$\mathbf{SiLU}(x) \approx \frac{x}{p} \sum_{j=1}^{p} \Phi\left(\frac{x}{\acute{\sigma}_j}\right)$$

and we can write the derivatives (with order $s \geq 1$) of **SiLU** as

$$\frac{\partial^s}{\partial u^s} \mathbf{SiLU}(x) = \frac{s}{p} \sum_{j=1}^{p} \frac{\partial^{s-1}}{\partial u^{s-1}} \Phi\left(\frac{x}{\acute{\sigma}_j}\right) + \frac{x}{p} \sum_{j=1}^{p} \frac{\partial^s}{\partial u^s} \Phi\left(\frac{x}{\acute{\sigma}_j}\right)$$

The rest of the derivation is very similar to that of **GELU**, so we only list the final results. With

$$\hat{\sigma}_j^2 = \tilde{\sigma}^2 + \acute{\sigma}_j^2$$

$$
\begin{aligned}
A_0 &= \frac{1}{p} \sum_{j=1}^{p} \frac{\tilde{\mu}}{2} \mathbf{erfc}\left(-\frac{\tilde{\mu}}{\sqrt{2\hat{\sigma}_j^2}}\right) + \frac{\tilde{\sigma}^2}{\hat{\sigma}_j} \varphi\left(\frac{\tilde{\mu}}{\hat{\sigma}_j}\right) \\[2mm]
A_1 &= \frac{1}{p} \sum_{j=1}^{p} \frac{1}{2} \mathbf{erfc}\left(-\frac{\tilde{\mu}}{\sqrt{2\hat{\sigma}_j^2}}\right) + \tilde{\mu} \frac{\acute{\sigma}_j^2}{\hat{\sigma}_j^2} \frac{1}{\hat{\sigma}_j} \varphi\left(\frac{\tilde{\mu}}{\hat{\sigma}_j}\right) \\[2mm]
A_2 &= \frac{1}{2!\,p} \sum_{j=1}^{p} \left[1 + \frac{\acute{\sigma}_j^2}{\hat{\sigma}_j^2} + \frac{\tilde{\mu}^2 \acute{\sigma}_j^2}{\hat{\sigma}_j^4}\right] \frac{1}{\hat{\sigma}_j} \varphi\left(\frac{\tilde{\mu}}{\hat{\sigma}_j}\right) \\[2mm]
A_3 &= -\frac{1}{3!\,p} \sum_{j=1}^{p} \left[\frac{\tilde{\mu}}{\hat{\sigma}_j^2} + \frac{3\tilde{\mu}\acute{\sigma}_j^2}{\hat{\sigma}_j^4} - \frac{\tilde{\mu}^3 \acute{\sigma}_j^2}{\hat{\sigma}_j^6}\right] \frac{1}{\hat{\sigma}_j} \varphi\left(\frac{\tilde{\mu}}{\hat{\sigma}_j}\right) \\[2mm]
A_4 &= \frac{1}{4!\,p} \sum_{j=1}^{p} \left[-\frac{1}{\hat{\sigma}_j^2} + \frac{\tilde{\mu}^2 - 3\acute{\sigma}_j^2}{\hat{\sigma}_j^4} + \frac{6\tilde{\mu}^2 \acute{\sigma}_j^2}{\hat{\sigma}_j^6} - \frac{\tilde{\mu}^4 \acute{\sigma}_j^2}{\hat{\sigma}_j^8}\right] \frac{1}{\hat{\sigma}_j} \varphi\left(\frac{\tilde{\mu}}{\hat{\sigma}_j}\right) \\[2mm]
\dots
\end{aligned}
\tag{32}
$$

and the general form of $A_s(s \geq 2)$ is

$$A_s(s \geq 2) = \frac{1}{s!\,p} \sum_{j=1}^{p} \left[\left(\frac{-1}{\sqrt{2\hat{\sigma}^2}}\right)^{s-2} \mathbf{H}_{s-2}\left(\frac{\tilde{\mu}}{\sqrt{2\hat{\sigma}^2}}\right) - \left(\frac{-1}{\sqrt{2\hat{\sigma}^2}}\right)^s \mathbf{H}_s\left(\frac{\tilde{\mu}}{\sqrt{2\hat{\sigma}^2}}\right)\right] \frac{1}{\hat{\sigma}} \varphi\left(\frac{\tilde{\mu}}{\hat{\sigma}}\right) \tag{33}$$

