# OpenReview forum: "A Stochastic Polynomial Expansion for Uncertainty Propagation through Networks"
_TMLR — Rejected by TMLR_

### Review · Reviewer_kWJ4 · 2024-11-18

**Summary Of Contributions:**

The authors propose a novel method of propagating (variational) multivariate Gaussians through neural network layers, including different non-linear activation functions (tanh, sigmoid, softplus, ReLU, LeakyReLU, GELU, SiLU). For this purpose, they propose the use of a pseudo-Taylor polynomial expansion, which describes the expected output of a function given some stochastic input. They then derive the corresponding expressions to evaluate the covariance given different activation functions and validate their results in some experiments: The authors compare the effectiveness of the approximation when higher order terms are included in the pseudo-Taylor expansion compared to Jacobian linearization. They also perform a 1D regression experiment and a series of trials using ResNets of increasing depth on CIFAR10, where Gaussian noise is added to input images.

**Audience:**

Yes

**Broader Impact Concerns:**

I do not foresee any ethical concerns with this work.

**Claims And Evidence:**

Yes

**Requested Changes:**

First of all, I would like to emphasize that I think this paper would be of great interest for the TMLR readership.
I do appreciate the authors' work here, and would only like to propose changes in order to make the paper more accessible and some of the claims more convincing.

* On multiple points in the paper, I noticed some issues with citations. It looked like multiple citations where performed using \cite{author1}, \cite{author2} instead of \citet{author1, author2}. Similarly (\cite{author1}) -> \citep{author2}) and using \citeauthor{author1} instead of author1 et al.
* I found some minor typos:
    * "multi-variate"  instead of "multivariate" at the bottom of page 4.
    *" Peterson et. al." in section 4, "networksPeterson" just below.
    * No space between citation Janson (1997) and Michaelowicz et al. (2011) towards the end of appendix A.5.2.
* Some appendix sections like A.3, A.4, A.5 are never referenced in the main text. I would argue that A.4 should maybe be merged with A.5.
* I would like to see a proper related work / background section, so that the proposed method can be contrasted more explicitly with mentioned Jacobian linearlization and stochastic linearization. In case space is an issue, I might propose to move Table 1 and 2 to an appendix section, and e.g. only include terms for tanh / sigmoid for illustrative purposes in the main text.
* I find Table 3 a bit unruly and hard to take in. I would suggest the following potential improvements:
    * Round numbers to two decimal points to reduce column width.
    * Separate into two different tables for KLdiv and Frob. Cov. so that related numbers are side by side and easier to compare.
    * Potentially move the comprehensive version of the table to the appendix, and only show results for one activation function.
    * Potentially show the results in line or bar plots instead, so that the relationship between KLdiv / Frob. Cov. as a function of network depth or input noise variance becomes more apparent.
* I think the paper would be stronger if the usefulness of the obtained uncertainty would be evaluated experimentally in a downstream task such as OOD or misclassification detection as well. The authors hint to this in the conclusion, mentioning that "[our method] can potentially be effectively applied in various domains, including adversarial training, Bayesian inference, and safety-critical applications".

**Strengths And Weaknesses:**

Strengths
-----------

* The authors propose a (to my knowledge) new way to evaluate the activation distributions in deep neural networks in closed-form, and supply a large amount of derivations to make this method applicable to many different activation functions.
* They provide sensible experiments to validate their method empirically.


Weaknesses
---------------

* The work is missing a dedicated related work section, only some related works are discussed superficially in the introduction.
* The usefulness of the obtained uncertainty is not directly validated in experiments.
* The paper could profit from slight improvements in citation, presentation and organization.

---

### Review · Reviewer_vA5F · 2024-12-02

**Summary Of Contributions:**

This paper proposed a novel method for propagating uncertainty from the input of a neural network to its output, based on a "stochastic polynomial expansion" of the nonlinearities in the network. The authors provide closed form expansions for the first four polynomial coefficients for seven commonly used activation functions, and provide a general methodology for computing additional coefficients for general non-linearities. The authors provide experimental evidence for the accuracy of their approximations based on 1D regression problems and Cifar-100.

**Audience:**

Yes

**Claims And Evidence:**

No

**Requested Changes:**

Though I list each clarity issue as a "strengthen" change, taken together I believe that they constitute a "critical" change that would be required to secure my recommendation.

1. [strengthen] Provide more explanation for Figure 2. This figure isn't referenced anywhere in the text. Furthermore, it is unclear to me what I am supposed to take away from this figure. Presumably the purpose to is compare the authors method with two baselines, but it isn't clear how to draw any conclusions from the comparison.
2. [strengthen] Clarify that Hernandez-Lobato and Adams are not using a Variational approximation, but are instead making use of the expectation propagation framework.
3. [strengthen] What is the meaning of the upper script $\circ$ and $s$ in $\Xi_{(s)}^{\circ s}$? How is $\Xi_{(s)}^{\circ s}$ different from $\Xi_{(s)}$?
4. [strengthen] In table 1, it isn't clear that the provided polynomials are approximate polynomials that depend on approximations that are specific to each of the activation functions. It also isn't clear which value $C_j$ takes in each case.
5. [critical] Similarly, there are no experimental results which aim to demonstrate the sensitivity of the proposed method to such approximations. How does performance degrade as the number of approximating terms are increased or decreased from 4, compared with a baseline of MC estimating (with many samples) the intractable integrals?
6. [critical] The experimental results are all in "toy" settings which fail to illustrate whether the method is useful in real world settings. In the 1D regression setting, it would be good to see experiments with "in-between uncertainty" (see "'In-Between' Uncertainty in Bayesian Neural Networks" by Foong et al.). It would also be good to see experiments which demonstrate that the uncertainty estimates produced by this method are useful for predictive tasks, for example on rotated-MNIST, corrupted-CIFAR, and OOD detection tasks (see "Can you trust your model's uncertainty? Evaluating predictive uncertainty under dataset shift" by Ovadia et al.). At the very least, it would be good to see how this method compares with MC-Dropout, MFVI, Deep Ensembles, and the method of Hernandez-Lobato and Adams on the UCI dataset benchmark from the latter's paper. The rotated-MNIST and corrupted-CIFAR experiments would be particularly valuable since they provide somewhat more realistic sources of noise than the added Gaussian noise in this paper's CIFAR experiments.
7. [strengthen] Please clarify how exactly this method is integrated into DVI?
8. [strengthen] Given the toy nature of the single DVI experiment in the paper, a statement like "The strong alignment between true and predicted intervals demonstrates PTPE's effectiveness ..." is too strong, please add more realistic experiments or weaken the statement.
9. [strengthen] Please add some context for the chosen input variances in table 3. It would be helpful to understand how relevant noise at this scale is to "real-world" experiments.

**Strengths And Weaknesses:**

## Strengths

* The paper tackles an interesting and challenging problem.
* The paper provides a novel solution to this problem, and in doing so provides approximated moments for several commonly used activation functions. These moments could be useful outside of this line of work.

## Weaknesses

* The paper is unclear in several places. In particular, the mathematical notation is not well described, making it difficult to follow proofs, lemmas, and derivations. Specific issues will be listed in the next section.
* The experimental evidence provided only covers very restricted / toy settings, which makes it unclear how well the method actually works in practice.
* The method is sold as being general, but it first requires that an activation function is approximated in such a way that the intractable integrals in the expectations tractable analytically. This means that it may be non-trivial to apply this method to an arbitrary activation function. Furthermore, the authors have not provided any sensitivity analyses to show how this approximation step impacts the overall accuracy of the method.

---

### Review · Reviewer_G3XD · 2024-12-13

**Summary Of Contributions:**

This work proposes a method for propagating Gaussian inputs through a non-linearity $f$ by local polynomial approximation. The authors focus on the setting where each co-ordinate of $f$ depends only on the corresponding co-ordinate of $X$, as in the non-linearity in a neural network, so that the derivatives of $f$ are all diagonal. They then use a truncated Taylor series approximation, at each step using the expectation of the derivative, and due to the diagonality, the cost is only $O(n^2)$ for length $n$ inputs.

They provide approximations to permit calculation of the expected derivatives of various widely used non-linearities, and explicit expressions for the expansion.

Experimentally, they show that the third order expansion performs empirically much better than existing first order expansions (local linearisations), in the sense that the propagated distribution is much closer to the true distribution, at least when the input variance is fairly large.

**Audience:**

Yes

**Claims And Evidence:**

Yes

**Requested Changes:**

* The use of ∘, both for element-wise multiplication and element-wise exponentiation may be widespread (?) but it's unfamiliar to me. I would suggest explicitly stating this early on to make the notation clearer.
* Similarly, I would expect $∇_X f(X)$ to be $n×n$; it would be good to explain exactly what you mean by this here.
* The cancellation of the even terms in Lemma 2 is not explained even in the proof in the appendix. It's clear enough to me after thinking about it for a moment but you ought to add a sentence explaining it.
* See also weaknesses above.

Minor typographic elements throughout:
There are many in section 1.
* end of page 1: not only measurement -> not only from measurement
* near end of page 2: references need commas or semicolons separating them (+ 'and' between the last two)
* same place: The work Shekhovstov -> The work of Shekhovstov
* same place: independent Gaussian input -> independent Gaussian inputs
* paragraph "contributions": introduce errors from ignoring higher order -> introduces errors from ignoring the higher order
* same place: to covariance calculation -> to the covariance calculation
* same place: introduce errors -> introduces errors
* same place: ignoring correlations' contributions to variance -> ignoring the correlations' contributions to the variance calulcation
* same place: If deterministic -> If a deterministic
* next para: and derived closed-form -> and derives closed-form
* last para of section 1: last two references should be separated by 'and', Often, deterministic -> Often, a deterministic

**Strengths And Weaknesses:**

Strengths:
* (major) The empirical evaluation is quite thorough, and the authors are generally quite clear about the contributions
* (major) I think the principles of the idea are fairly clearly presented, although I make a few suggestions for changes in the next section.

Weaknesses
* (major) It's not really clear to me from the main text of the paper exactly how this method differs from/compares to general Gaussian filtering methods like cubature Kalman filter (see e,g, chapter 6 of Bayesian filtering and smoothing by Simo Sarkka). It would be helpful if the authors would discuss this clearly, and the paper would really shine if an explicit comparison was included (so that readers can judge how all these different methods compare). Especially since approximations are needed to calculate the expectations of the gradients of some of the nonlinearities.
* (major) There are some unanswered questions from the experiments. In particular, it's a little surprising to me that the approximation gets a little worse with higher order approximations in some cases. I understand that this is a 'low-noise' case, and one would expect stochastic linearisation to do okay in lower noise, but it's not so clear why higher order approximations would be worse. Some insight into this would be useful, as would error bars on the values (e.g., are the higher order approximations worse or are they actually all about the same?)
* (minor) It's not really clear how the experiments relate to real-world settings: for example, the results of 3.2 involve corrupting inputs with different levels of noise and checking how well uncertainty is propagated, but it's not so clear what the calibration error would be like in a real setting. The authors could address this by, e.g. adding a more representative real-world experiment, or by suitable discussion.

---

### Decision · Action_Editor_iyoG · 2025-02-10

**Recommendation:** Reject

**Comment:**

Reviewers recommend rejection for this paper. Their major questions for this submission are:
1. Clarification on the method regarding related work (e.g., approximations done in Kalman filtering), and not enough literature review.
2. Clarity on the math presentations.
3. Experiments on real-world datasets are missing -- only with experiments on neural networks showing approximation error to Monte Carlo estimates.

No author reply is provided.

I recommend the authors to check the reviewers' comments and consider improvements accordingly.

**Audience:**

Machine learning researchers interested in uncertainty quantification.

**Claims And Evidence:**

This work considers uncertainty quantification in neural network by propagating Gaussian distributions through the network. The main contribution is the truncated polynomial approximations to a list of commonly used activation functions to enable accurate approximation of the mean and variance of transformed Gaussian. Experiments have validated the efficacy of the approximations on simulated benchmarks.